# Mating can initiate stable RNA silencing that overcomes epigenetic recovery

Sindhuja Devanapally [1,2], Pravrutha Raman [1,2], Mary Chey [1], Samual Allgood [1], Farida Ettefa[1], Maïgane Diop [1], Yixin Lin [1], Yongyi E. Cho [1] & Antony M. Jose [1✉]

Stable epigenetic changes appear uncommon, suggesting that changes typically dissipate or are repaired. Changes that stably alter gene expression across generations presumably require particular conditions that are currently unknown. Here we report that a minimal combination of *cis*-regulatory sequences can support permanent RNA silencing of a single-copy transgene and its derivatives in *C. elegans* simply upon mating. Mating disrupts competing RNA-based mechanisms to initiate silencing that can last for >300 generations. This stable silencing requires components of the small RNA pathway and can silence homologous sequences in *trans*. While animals do not recover from mating-induced silencing, they often recover from and become resistant to *trans* silencing. Recovery is also observed in most cases when double-stranded RNA is used to silence the same coding sequence in different regulatory contexts that drive germline expression. Therefore, we propose that regulatory features can evolve to oppose permanent and potentially maladaptive responses to transient change.

[1] University of Maryland, College Park, MD, USA. [2] These authors contributed equally: Sindhuja Devanapally, Pravrutha Raman. ✉email: amjose@umd.edu

When organisms reproduce by building a near copy of themselves, they recreate the information needed for making another copy. This heritable information is stored as the genome sequence and as particular spatial arrangements of regulators within each new generation[1–3]. Rare mutations in genome sequence that result from failed DNA repair are transmitted across generations through DNA replication during each cell division. Unlike such genetic changes, epigenetic changes, which do not alter genome sequence, can result in three possible outcomes: passive dilution, active repair through negative feedback, or active maintenance through positive feedback. Therefore, both the mechanisms that detect change and the associated regulatory contexts are relevant for the persistence of epigenetic changes.

Stable epigenetic changes that last for hundreds of generations have been observed in a variety of systems. In every case, they are characterized by mechanisms that include positive feedback for copying or amplifying the change. For example, in *Paramecium aurelia*, where changes in the cortical arrangement of cilia can be stable, new rows of cilia are made using previous rows as templates[4]. In wild *Saccharomyces*, where changes in protein folding can persist for many generations, the folded structures of prion proteins template the folding of newly made proteins[5]. In *Cryptococcus neoformans*, where changes in ancestral DNA methylation can potentially persist for millions of years, methyltransferases copy methylation patterns upon DNA replication[6]. In every case, positive feedback ensures that ancestral epigenetic changes are not diluted across generations as cells divide.

Positive feedback alone, however, does not guarantee the stability of an epigenetic change across generations. For example, although the presence of RNA amplification correlates with reported cases of persistent RNA silencing, most induced silencing dissipates within a few generations (reviewed in ref.[7]). In *Caenorhabditis elegans*, RNA-dependent RNA polymerases (RdRPs) are used for the small RNA-guided production of additional small RNAs[8] that are complementary to terminally modified mRNA fragments[9]. This cycle of small RNA production can act across generations, leading to effects that last for varying numbers of generations (Supplementary Table 1). However, the mere presence of small RNAs (Supplementary Table 1) or terminally modified mRNA fragments[9] does not result in indefinite RNA silencing. Changes lasting for a few generations that cannot be explained by direct parental effects have been considered transgenerational[10]. Such temporary transgenerational changes could be qualitatively distinct from induced changes that are stable for hundreds of generations. These considerations suggest that other currently unknown factors that recruit or enhance positive feedback mechanisms are crucial for stable epigenetic changes.

Here we introduce mating as a simple approach to reproducibly initiate RNA silencing of a single-copy transgene that can last for hundreds of generations. A minimal combination of *cis*-regulatory sequences from this transgene can support such stable change within the *C. elegans* germline. Genes that share subsets of these regulatory sequences can be silenced for a few generations, but subsequently recover from and even become resistant to some forms of RNA silencing. Thus, our results establish a paradigm for analyzing the regulatory differences that determine persistent epigenetic change versus epigenetic recovery.

## Results

**Mating can disrupt gene expression by initiating piRNA-mediated silencing.** We serendipitously discovered that a previously generated two-gene operon (the single-copy transgene *oxSi487*, ref.[11] Fig. 1a and Supplementary Fig. 1a) has an exceptional capacity for retaining changes in gene expression for many generations. This transgene referred to here as *T* encodes a bicistronic operon that expresses *mCherry* and *gfp* in the germline, presumably as one pre-mRNA transcript before being spliced into mature mRNAs (Fig. 1b and Supplementary Fig. 1b). While progeny that inherit *T* maternally showed uniform mCherry and GFP expression, progeny that inherit *T* paternally showed loss of expression (Fig. 1c, d, left), despite stable expression of *T* within the male parents (Fig. 1a). Mating alone is not sufficient to cause silencing because when both parents expressed *T*, all descendants showed stable expression (Fig. 1d, left, Supplementary Fig. 1c). Hemizygosity alone is not sufficient to cause silencing because all hemizygous descendants generated from a cross of wild-type males with hermaphrodites that express *T* showed stable expression (Supplementary Fig. 1d). Hermaphrodite sperm were not necessary for this phenomenon because cross progeny of feminized animals, which cannot self-fertilize, mated with transgenic males showed silencing (Supplementary Fig. 1e). We refer to this silencing that is initiated upon mating males with *T* to hermaphrodites or females without *T* as mating-induced silencing because it appears to be distinct from previously reported epigenetic silencing phenomena (Supplementary Table 2). Although the extent of the observed mating-induced silencing was variable in progeny, it was initiated in every cross where *T* was inherited only paternally (Fig. 1d, right). Initiation was extremely reliable because it was observed in >1500 animals from each one of >140 independent crosses in wild-type, *dpy*- or *unc*-marked genetic backgrounds. In every comparison, precisely the same markers were used to control for genetic background of animals being compared. The extent of mating-induced silencing ('dim' and 'off' animals), however, varied from 68 to 100% of cross progeny scored depending on the context of different genetic markers. Since the reasons for this variability are currently unclear, we did not make strong inferences from small variations in this study. Mating-induced silencing was not observed with other genes, including those sharing extensive sequence identity with *T* (Supplementary Fig. 2a, b). We also did not detect any significant differences in abundance or subcellular localization of RNA transcripts of *T* and susceptible variants of *T* (Supplementary Fig. 3a) compared to those of genes not susceptible to mating-induced silencing (Supplementary Fig. 2c–e). Thus, mating-induced silencing can be initiated reproducibly at the population level and the susceptibility of *T* to silencing without the addition of external triggers provides a reliable paradigm for inducing and analyzing the stability of epigenetic change.

To discover the parts of *T* that are required for its susceptibility to mating-induced silencing, we systematically modified sequence features of *T* (Supplementary Fig. 3a). All tested variants of *T* were susceptible to mating-induced silencing (Fig. 1e and Supplementary Fig. 1f–h), including *Tcherry*, a minimal variant comprising *Pmex-5* promoter driving expression of *mCherry* with a *cye-1* 3′ UTR (Fig. 1e and Supplementary Fig. 3a). Therefore, operon structure, histone sequences, co-transformation marker (*C. briggsae unc-119(+)*), and the method of genomic integration are not sufficient to explain susceptibility to mating-induced silencing. Germline gene expression in *C. elegans* can depend on 3′ UTRs[12,13] and genomic position[14]. Neither altering the 3′ UTR nor changing the genomic position eliminated susceptibility of *Tcherry* to mating-induced silencing (Fig. 1e, Supplementary Figs. 1i and 3a). However, when *mCherry* sequence from *Tcherry* was fused to the endogenous *mex-5* gene within the context of native regulatory features, *mCherry* became resistant to mating-induced silencing (Fig. 1a, e). Resistance to silencing cannot be attributed merely to presence of endogenous sequences because *T* was susceptible despite the presence of histone *h2b* coding

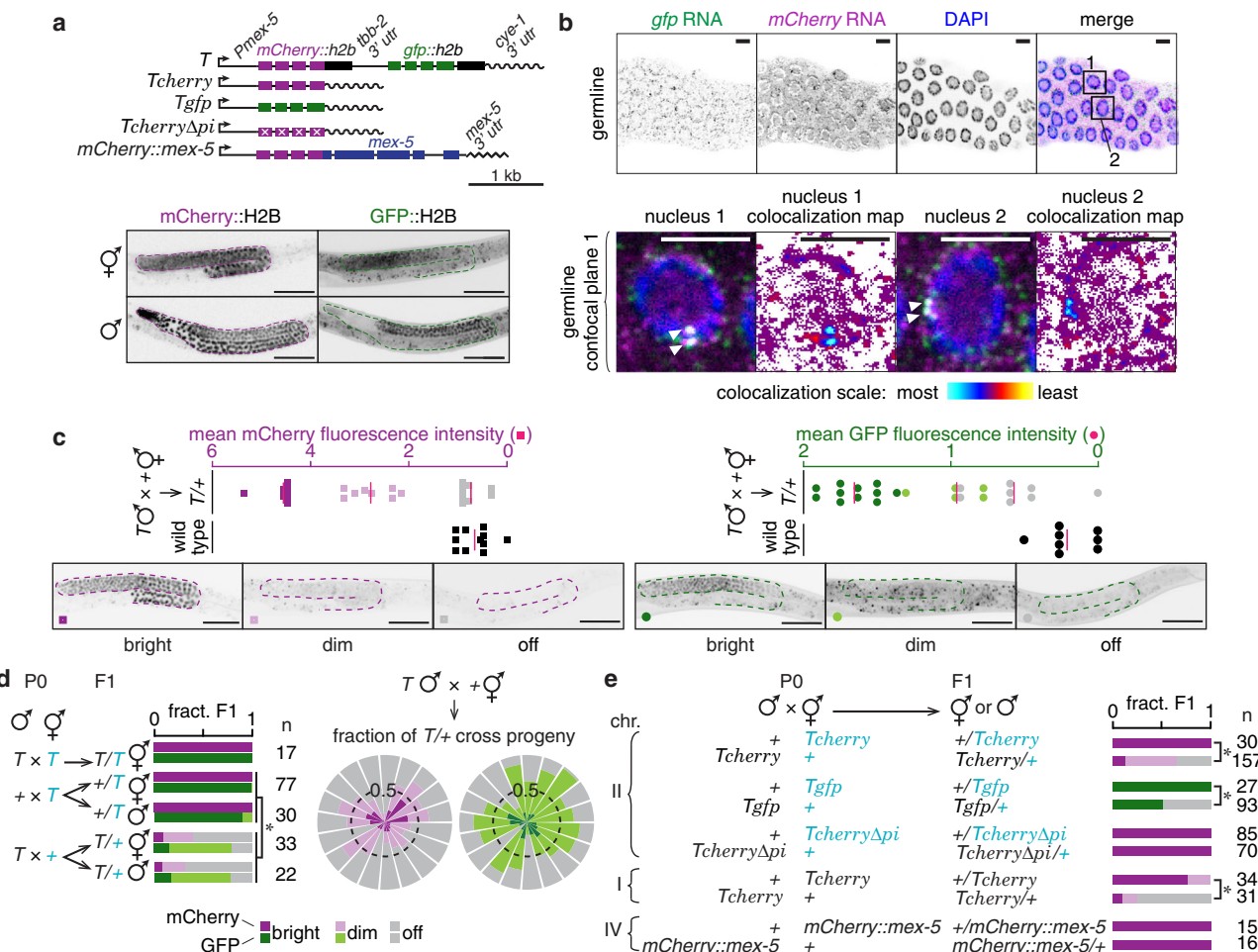

**Fig. 1 Mating can disrupt gene expression by initiating RNA silencing. a** Schematics of the single-copy transgene *Pmex-5::mCherry::h2b::tbb-2 3'utr::gpd-2 operon::gfp::h2b::cye-1 3' utr* referred to as *T*, of independently generated minimal variants expressing only *mCherry* (*Tcherry*), *gfp* (*Tgfp*) or *mCherry* lacking piRNA-binding sites (*TcherryΔpi*) and of *mCherry* fused to endogenous *mex-5* coding sequence (*mCherry::mex-5*) are depicted (top). Germlines (dotted outline) of representative L4-staged hermaphrodites and males showing mCherry (magenta) or GFP (green) expression from *T* are indicated (bottom). **b** Single-molecule fluorescence in situ hybridization (smFISH) of *mCherry* and *gfp* in dissected gonads of animals expressing *T* reveals that *mCherry* RNA and *gfp* RNA colocalize as one or two spots (white arrowheads) within nuclei. Representative confocal plane (germline) imaged from a dissected gonad. Additional images are in Supplementary Fig. 1b. Colocalization heat map represents the extent of overlap between pixels corresponding to *mCherry* RNA and *gfp* RNA. **c** Quantification (top) and representative images (bottom) of the germline (magenta outline and green outline) of hemizygous animals (*T/+*) scored as having bright, dim, or not detectable (off) mCherry (left) or GFP (right) fluorescence. Average normalized fluorescence (red bar) within the germline was calculated for 11 bright, 5–8 dim, 8 off (gray), and 7 wild-type (black) L4-staged hermaphrodites. **d** Cross progeny males and hermaphrodites that inherited *T* from one or both parents were scored for expression of mCherry and GFP from *T* (left). Rose plot of independent repeats of mating-induced silencing of *T* (right). Each segment (mCherry, left and GFP, right) represents independent trials of one to four biological replicates and includes data from experiments depicted in other figures within the manuscript (total n = 561 animals). Dashed circles indicate half the fraction of animals scored. **e** Animals expressing *T, Tcherry, Tgfp, TcherryΔpi* or *mCherry* fused to the endogenous *mex-5* coding sequence (*mCherry::mex-5*) were mated with non-transgenic animals and resulting cross progeny were scored as having bright (magenta or green), dim (pink or light green), or not detectable (off, gray) levels of mCherry or GFP fluorescence. Number next to curly brackets refers to the chromosome on which each gene is present. All scored cross progeny were hermaphrodites except in the case of animals with *Tcherry* on chromosome I, where males were scored to ensure that cross-progeny were scored. In all figures, homozygous genotypes are indicated as a single character for simplicity—for example, '*T*' represents homozygous *T/T* animals, '+' represents non-transgenic or wild-type (+/+) animals etc. Also see Supplementary Figs. 1–7. Asterisks indicate P < 0.05 using χ² test. Chromosomes with a *dpy* marker (blue font) and numbers of animals scored (n) are indicated. Scale bars, 50 μm (**a, c**) and 5 μm (**b**). Source data are provided as a Source Data file.

regions. Lastly, other transgenes with homologous intron sequences (*Dendra2::H2B* in Supplementary Fig. 2a) or *mex-5* promoter (*mCherry^var2::mex-5* in Supplementary Fig. 2a) were not susceptible to mating-induced silencing. Thus, regulatory features that contribute to *Tcherry* expression (*cis*-regulatory sequences, intranuclear localization of DNA, chromatin neighborhood, etc.) are sufficient to support change in gene expression upon mating.

To examine if unequal partitioning of parental factors could cause preferential mating-induced silencing in early progeny as observed during RNA interference (RNAi)[15,16], we separately measured silencing in four successive cohorts of progeny (Supplementary Fig. 1j). Proportions of animals that showed silencing were comparable in each cohort, ruling out such systematic bias. The variation in mCherry and GFP fluorescence was correlated in most individual F1 animals (Supplementary Fig. 1k), suggesting that silencing occurs either on unspliced pre-

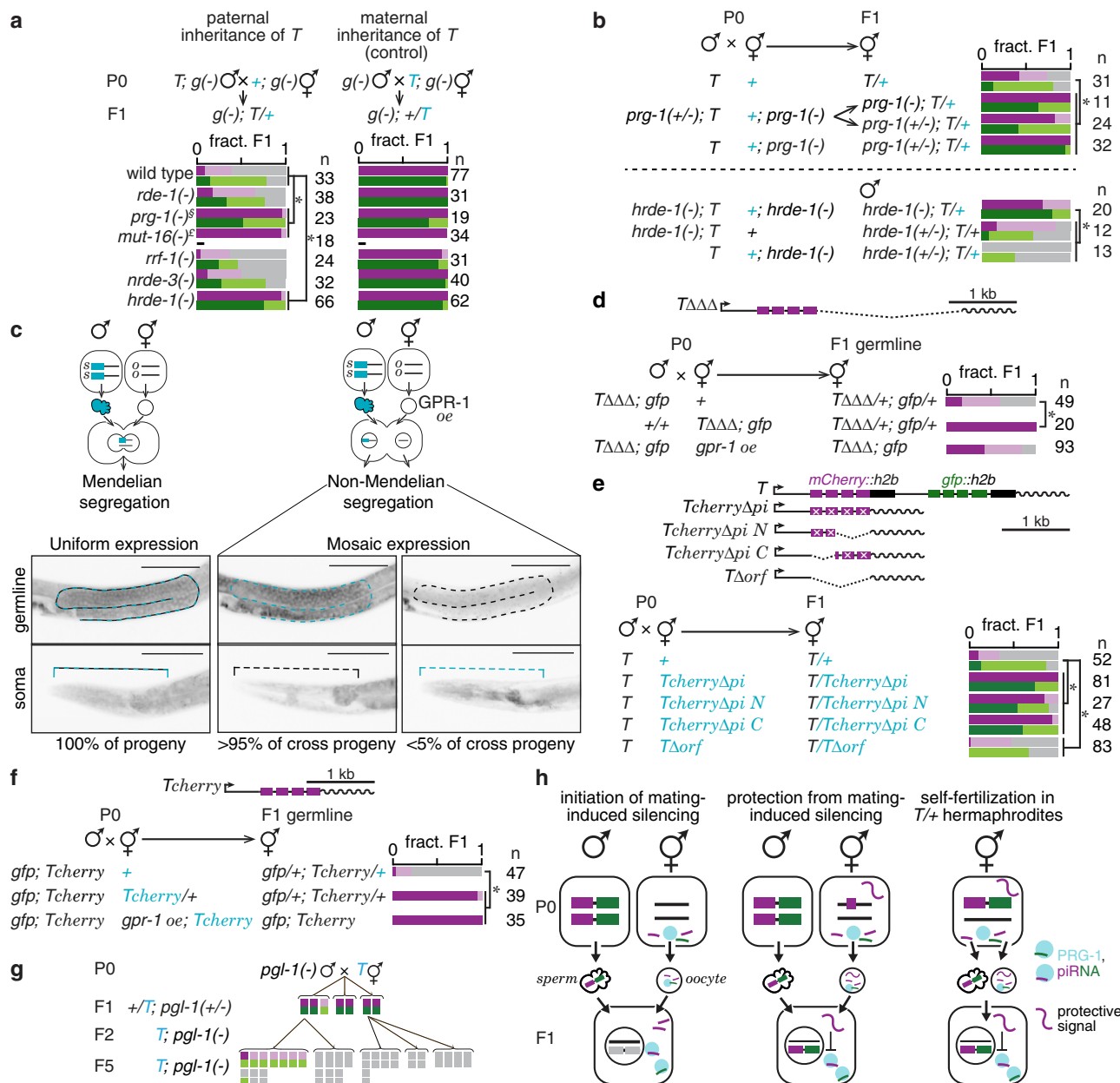

mRNA or simultaneously on both *mCherry* and *gfp* mRNA during or after pre-mRNA splicing.

Examining known RNA silencing factors (Supplementary Fig. 3b; refs. [9,17].) revealed that mating-induced silencing required the primary Argonaute PRG-1, mutator protein MUT-16, and the secondary Argonaute HRDE-1 (Fig. 2a), distinguishing it from silencing by feeding RNAi, which was PRG-1-independent (Supplementary Fig. 3c). Four observations support an intergenerational mechanism for the initiation of mating-induced silencing using PRG-1-bound small RNAs called piRNAs. One, reduction in protein fluorescence from *T* was accompanied by reduction in RNA levels in silenced progeny (Supplementary Fig. 4a, b). Two, removal of predicted piRNA sites[18] in *mCherry* (*TcherryΔpi*) eliminated mating-induced silencing (Fig. 1e and Supplementary Fig. 3d). Three, maternal absence of PRG-1 and zygotic absence of HRDE-1 prevented initiation of silencing (Fig. 2b). Four, preventing pronuclear fusion in progeny using maternal overexpression of the G-protein regulator GPR-1 (Fig. 2c, d; refs.[19,20]; see Methods) still resulted in silencing, indicating that initiation is independent of maternal chromatin in

the germline of progeny. Because PRG-1 loss abolished mating-induced silencing of *gfp* in *T* (Fig. 2a, b) and because *Tgfp* (*Pmex-5::gfp::cye-1 3' UTR*) was also silenced by mating, there is a potential for endogenous piRNAs complementary to *gfp*[18] to trigger mating-induced silencing of *gfp* independent of *mCherry*. Together, these results suggest that maternal PRG-1-bound piRNAs trigger production of secondary small RNAs using transcripts from *T* (Fig. 2h, left) and MUT-16-dependent perinuclear mutator foci[21], which then bind HRDE-1 and cause silencing in progeny.

In summary, mating can disrupt the expression of a set of single-copy transgenes and cause transgenerational RNA silencing. This stable RNA silencing could result from the activation/gain of mechanisms that promote silencing or the repression/loss of mechanisms that prevent silencing, or both.

**Homologous maternal transcripts protect against initiation of mating-induced silencing.** Initiation of mating-induced silencing of paternally inherited *T* could be prevented by maternal

**Fig. 2 Requirements for initiation of and protection from mating-induced silencing. a** Mating-induced silencing was initiated as in Fig. 1d in a wild-type or in different mutant (*g*(-)) backgrounds (left) and silencing in resulting cross progeny were compared with that of the same genotypes from control crosses (right). Wild-type crosses shown here are the same as in Fig. 1d. An additional wild-type cross with a different visible marker (not depicted, but showed mating-induced silencing—mCherry: bright = 5, dim = 6, off = 25 and GFP: bright = 7, dim = 12, off = 17 in F1 progeny) was performed for comparison with the *rde-1*(-) cross on the right. Use of *prg-1*(−/+) parent males owing to the poor mating by *prg-1*(-) parent males is indicated (§). Requirement of *mut-16* in initiation of silencing was examined by scoring only mCherry fluorescence, but not GFP fluorescence, in male cross progeny (£). **b** Requirement of *prg-1* and *hrde-1* in initiation was tested by mating parents mutant for either of these genes and scoring cross progeny. **c** Scheme to test effect of *gpr-1* overexpression: *gtbp-1::gfp* (blue) males mated with wild-type hermaphrodites (left) or with hermaphrodites overexpressing *gpr-1* in the germline (*gpr-1 oe*, right). *s* and *o* label DNA inherited through sperm and oocyte respectively. Representative images show differences in segregation of *gtbp-1::gfp* in the germline (top) and the head (soma, bottom) in cross progeny. Colored outlines and brackets show the parental origin of germline or pharynx. **d** Animals expressing *TΔΔΔ* and *Pgtbp-1::gtbp-1::gfp* marker (*gfp*) were mated with either non-transgenic animals or animals overexpressing *gpr-1* (*gpr-1 oe*). Fluorescence from mCherry in the germline of cross progeny was scored. Gene structures are also depicted here (top) and in other panels. **e** *T* males were mated with hermaphrodites expressing variants of *TcherryΔpi* and progeny with paternally inherited *T* were scored. **f** Males expressing *Pgtbp-1::gtbp-1::gfp* marker (*gfp*) and *Tcherry* were mated with hermaphrodites that expressed *Tcherry* in a wild-type or *gpr-1* overexpression (*gpr-1 oe*) background and fluorescence of paternally inherited *Tcherry* was scored in cross progeny. **g** *T* animals were mated with *pgl-1* mutants and expression of *T* was assessed in hemizygous cross progeny and in homozygous descendants. Each vertical pair of boxes represents fluorescence intensity of mCherry and GFP within the same animal. **h** Schematics depict inferences from mating-induced silencing: left, when *T* males are mated with hermaphrodites lacking *T*, cross progeny are silenced by PRG-1 inherited through the oocyte using piRNAs targeting *mCherry* and *gfp* in *T*; middle, when *T* males are mated with hermaphrodites expressing a fragment of *T*, even with piRNA target sites mutated (e.g. *TcherryΔpi N*), cross progeny are protected from silencing initiated by PRG-1 inherited through the oocyte; right, when hemizygous hermaphrodites self-fertilize using transgenic sperm carrying *T* and oocytes that lack *T* but carry piRNAs targeting *T*, self-progeny remain unsilenced possibly due to a protective signal, likely RNAs, derived from parental *T* transmitted through the oocyte into progeny. Chromosomes with a *dpy* marker (blue font), number (*n*) of animals scored (**a**, **b**, **d**–**g**) and scale bars, 50 μm, (**c**) are indicated. Scoring of fluorescence is as in Fig. 1. Also see Supplementary Figs. 1, 3, 4. Asterisks indicate $P < 0.05$ using $\chi^2$ test. Also see 'Genetic Crosses' under Methods. Source data are provided as a Source Data file.

expression of *T* (Fig. 2e), suggesting that a signal derived from maternal *T* can protect paternal *T* from silencing. Consistently, the protective signal mapped to a ~3.2 Mb region that includes *T* (Supplementary Fig. 5a). This ability to protect was also largely retained by variants of *T* containing *mCherry* (Fig. 2e, Supplementary Figs. 3a and 5b, c). Maternal presence of *mCherry*, even as a hemizygous single copy could protect both *mCherry* and *gfp* in more progeny compared to maternal presence of two copies of *gfp* (Supplementary Fig. 5b, c). This protective signal could explain why hemizygous self-progeny of hemizygous hermaphrodites showed stable expression of *T* for multiple generations even if *T* inherited through self-sperm is capable of being silenced (Fig. 2h, right, Supplementary Fig. 1d). In each generation of hemizygous hermaphrodites, the transgene is expected to be inherited through self-sperm 50% of the time (Supplementary Fig. 1d) and a maternal protective signal could be required for expression of *T* inherited through self-sperm. Therefore, either a protective signal inherited through oocytes licenses expression of *T* inherited through self-sperm in each generation or *T* inherited through self-sperm is not susceptible regardless of whether there is a protective signal inherited through the oocytes. Once paternally inherited *T* was protected, expression of *T* was stably maintained in descendants generated by self-fertilization (Supplementary Fig. 5d). Nevertheless, protected cross progeny remained susceptible to initiation similar to unsilenced progeny that escaped initiation of mating-induced silencing (Supplementary Fig. 5e, f). Because maternally expressed variants of *T* could confer protection despite nonsense mutations or deletions that disrupted the coding sequence (Supplementary Figs. 3a and 5b), the protective signal could be derived from parts of *T*. Consistently, *TcherryΔpi* sequences showed the strongest level of protection despite the inability of their transcripts to bind piRNAs, even when the N- or C-terminal sections of *TcherryΔpi* coding sequence were deleted (Fig. 2e, *TcherryΔpi N/C*). Therefore, protection cannot be explained by a simple model whereby complementary maternal *mCherry* sequences compete away maternal piRNAs. Protection was weak when only the last exon of *TcherryΔpi* was used (Supplementary Fig. 5b, *TcherryΔpi exon 4*) and was completely abolished when the entire open reading

frame was deleted (Fig. 2e, *TΔorf*). Other genes that share the same *mCherry* protein sequence or additional DNA sequences identical to regions of *T* could not protect *T* (Supplementary Fig. 5g, h). The protective signal did not require interactions between homologous chromosomes because *TcherryΔpi* on chromosome II could protect *Tcherry* on chromosome I from mating-induced silencing (Supplementary Fig. 5i). Lastly, preventing fusion of zygotic pronuclei still resulted in protection of paternal *T* (Fig. 2f). Collectively, these observations suggest that protection relies on a diffusible sequence-specific signal, likely maternally inherited transcript(s).

We noted that different *TcherryΔpi* variants appeared to protect in proportion to their coding-sequence lengths regardless of the number of mutated piRNA target sites (Supplementary Fig. 5j; also see Fig. 2e and Supplementary Fig. 5b). This observation suggests that either variants of *T* produce different amounts of the protective signal or that maternally inherited transcripts themselves protect by titrating away silencing small RNAs made against *T* triggered downstream of piRNA-binding in progeny (Supplementary Fig. 5j, k). The Argonaute CSR-1 has been proposed to play a role in promoting the expression of germline genes[22,23] and in the prevention or reversal of transgene silencing in the germline[24,25]. CSR-1 has also been proposed to regulate spermiogenesis and oogenesis[23], to silence sperm-specific transcripts in coordination with germ granules[26], and to tune the levels of germline transcripts[27]. These diverse roles make effects caused by the loss of CSR-1 difficult to interpret. Furthermore, the embryonic lethality caused by chromosome segregation defects in *csr-1* mutants[28] makes rigorous analyses across generations challenging. Nevertheless, we examined a component of the CSR-1 pathway that interacts with these small RNAs but lacks the confounding developmental defects. Unlike CSR-1, removal of the uridylyltransferase CDE-1 that uridylates CSR-1-associated small RNAs causes fewer pleiotropic effects[28,29]. CDE-1 loss did not abolish protection (Supplementary Fig. 5l). Although additional experiments are needed to identify the molecular machinery that mediates protection from mating-induced silencing, the ability of *TcherryΔpi* variants to protect both mRNAs from *T* suggests that the derived maternal signals

from *T* engage more complex mechanisms that license expression within the germline[30]. Protection of germline transcripts from piRNA-mediated silencing can occur within phase-separated condensates called P granules, which when disrupted can cause mis-regulation and aberrant distribution of some endogenous transcripts[31,32]. We tested for potentially similar mis-regulation of transcripts from *T* and observed complete silencing in some animals upon loss of the P granule component PGL-1 without the need for mating (Fig. 2g). This observation suggests that stable transgenerational expression of *T* likely reflects reliable recognition of transcripts from *T* within P granules as part of 'self' in every generation, according to some current models.

Thus, mating disrupts competing RNA-based mechanisms that regulate expression to initiate silencing (Fig. 2h) and maternal transcripts with partial homology are sufficient to oppose silencing by piRNAs. Protection by maternal transcripts explains the directionality of mating required for silencing and, in hindsight, also suggests explanations for the situations where we did not observe silencing (Supplementary Fig. 1l).

**Silencing induced by mating is actively maintained for >300 generations.** Once the expression state of *T* was established in cross progeny after mating, the expression state remained similar in subsequent generations (Fig. 3a and Supplementary Fig. 5m). Descendants of silenced F2 animals stayed silenced in 100% of animals in each tested generation for more than 300 generations without additional selection (Fig. 3b, c, Supplementary Figs. 5n and 6a, b). We refer to animals carrying a stably silenced copy of the transgene or its variants obtained by mating-induced silencing with an *i* (e.g. *iT*, where *i* stands for inactive) in the remainder of the paper. Consistent with transgenerational RNA silencing, *iT* animals showed a ~30- to 37-fold decrease in mRNA and ~4- to 6-fold decrease in pre-mRNA (Supplementary Fig. 6c). Transgenerational silencing could be detected even with variants of *T* that include a minimal coding sequence (Supplementary Fig. 5o, p), suggesting that additional sequence features are not needed for stable heritable silencing. Silencing triggered by piRNAs can last for many generations and be associated with repressive chromatin modifications[22,33–36]. Among RNA silencing and chromatin factors (Supplementary Fig. 3b), transgenerational stability of mating-induced silencing required the nuclear Argonaute HRDE-1, the nucleotidyltransferase RDE-3/MUT-2, and the intrinsically disordered protein MUT-16 even after 250 generations of silencing (Fig. 3c and Supplementary Fig. 6d). MUT-16 and RDE-3/MUT-2 are present in perinuclear foci where they promote the production of secondary small RNAs by RdRPs RRF-1 and EGO-1[21]. We examined animals that lack RRF-1 alone and animals that in addition lack zygotic EGO-1 (because animals that lack both maternal and zygotic EGO-1 are sterile[37,38]). Despite the potential for maternal rescue of *ego-1*, there was recovery of weak mCherry and GFP expression in *rrf-1* (-) *ego-1*(-) double mutants but not in *rrf-1*(-) single mutants (Fig. 3c), implicating these RdRPs in maintaining silencing in every generation. Similarly, only some *hrde-1*(-) progeny of *hrde-1*(+/−) animals showed expression, potentially due to maternal rescue, but all *hrde-1*(-) progeny in the next generation showed expression (Supplementary Fig. 6d). Thus, transgenerational silencing of *T* reflects active establishment of silencing by secondary small RNAs in every generation for hundreds of generations rather than passive loss of gene expression through DNA mutation (e.g., as occurs during repeat-induced point mutation in *Neurospora*[39]). Once expression was recovered in *hrde-1* mutants, restoring wild-type HRDE-1 did not re-establish silencing (Supplementary Fig. 6e), indicating permanent loss of silencing signals. HRDE-1-bound small RNAs can recognize nascent

transcripts and recruit chromatin modifiers to establish repressive histone modifications (e.g., H3K9me3) at target genes[22,40]. Neither the histone methyltransferases MET-2[41,42] or SET-32[42,43] nor the chromodomain protein HERI-1[44] was required for silencing (Fig. 3c). Descendants from a lineage that experienced >250 generations of silencing showed no significant changes in H3K9 methylation (Supplementary Fig. 6f, g). While transgenerational epigenetic inheritance (TEI) induced upon mating may be associated with additional unexamined molecular changes, reduction in RNA levels without any associated chromatin modifications is sufficient to explain maintenance (Fig. 3c, model).

**A signal associated with stable RNA silencing can enable *trans* silencing of homologous sequences.** Continued requirement of HRDE-1 to maintain stable silencing indicates that RNA silencing is likely associated with production of new small RNAs in every generation driven by small RNAs and/or template RNAs inherited from the previous generation. This positive feedback acting across generations could affect the expression of homologous genes because small RNAs can diffuse and encounter other complementary sequences, potentially initiating silencing at these new targets (Fig. 4a). To test this possibility, we examined whether stable silencing of *iT* has any *trans* effects on other homologous genes. We found that *iT* transmitted through one gamete could silence *T* inherited from the other gamete, regardless of the number of generations for which *iT* remained inactive (Fig. 4b and Supplementary Fig. 7a). Furthermore, presence of *iT* in any one parent was sufficient to cause significant silencing in progeny that inherited *T* only from the other, unsilenced parent (Fig. 4c). This *trans* silencing signal either is or relies on HRDE-1-dependent small RNAs because it is mostly eliminated upon loss of zygotic HRDE-1 (Fig. 4d). While meiosis is completed in sperm before fertilization, it is stalled at prophase I in oocytes until fertilization[45]. Nevertheless, oocyte meiosis is completed early in the one-cell zygote such that only a haploid genome is present in the oocyte pronucleus when it meets the sperm pronucleus. By preventing fusion of the haploid nuclei, we observed that direct interaction between parental chromatin was dispensable for *trans* silencing to occur (Fig. 4e), suggesting that *trans* silencing relies on a signal that is separable from DNA. This DNA-independent signal when transmitted through sperm must have separated from DNA in the male germline but when transmitted through oocytes can separate from DNA either in the hermaphrodite germline or in the embryo. Yet, this separable signal was not detectably inherited for more than one generation independent of *iT*, suggesting that it requires parental presence of *iT* for production (Supplementary Fig. 7b). Once *T* was silenced in *trans* by *iT*, the newly silenced copy of *T* remained silenced across generations, even when propagated by selfing without a copy of the ancestral *iT* (Fig. 4f and Supplementary Fig. 7c). Thus, silencing relies on a signal that is maternally deposited in every generation. This heritable silencing signal could be either HRDE-1-dependent small RNAs or downstream effectors made zygotically in response to a different intergenerational signal.

We examined the potential spread of a silencing signal associated with *iT* to sequences at other genomic positions. Genes sharing coding sequence identity, but not genes with only intronic or protein sequence identity, were silenced within the germline by *iT* in *trans* (Fig. 4g and Supplementary Fig. 7d). *Trans* silencing could only be detected with a stably established *iT* but not simultaneously with initiation of mating-induced silencing of *T* (Supplementary Fig. 7e), suggesting that initiation and maintenance of mating-induced silencing are either quantitatively distinct (e.g., different amounts of small RNAs) or

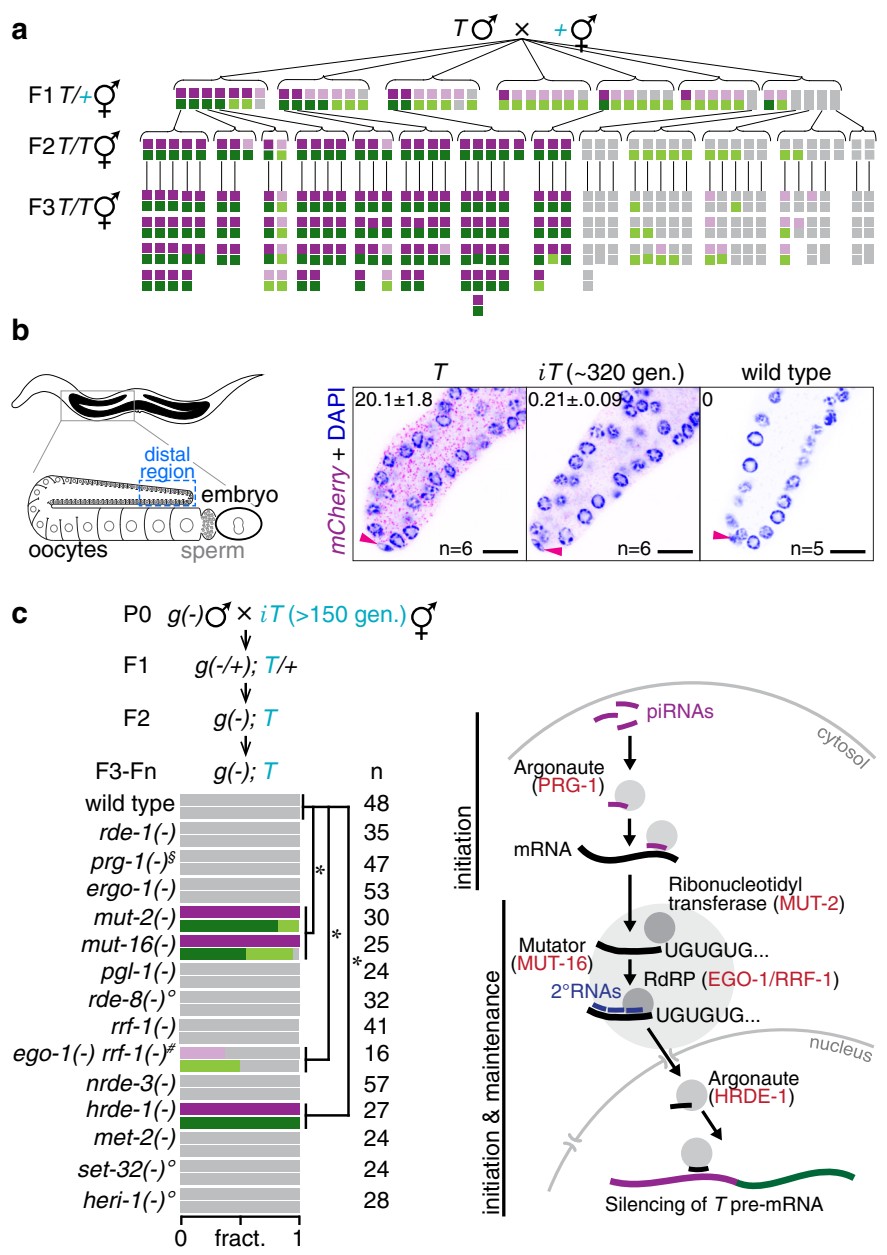

**Fig. 3 Mating-induced silencing is actively maintained for >300 generations. a** Mating-induced silencing was initiated and silencing was scored in cross progeny and their descendants. Each vertical pair of boxes represents fluorescence intensity of mCherry and GFP within the same animal (bright, dim, and off, as in Fig. 1c). **b** Maintenance of mating-induced silencing was measured by smFISH against *mCherry* exonic RNA in indicated distal region of dissected gonads of adult *T*, *iT* (silenced for ~320 generations) or wild-type animals. Pink arrowheads indicate the nucleus of the distal tip cell. Animals with median values of fluorescence or RNA signal in the distal region are shown in representative images. smFISH probes used are depicted in Supplementary Fig. 2d. Merged (DAPI + mCherry RNA smFISH) images shown here are also shown in Supplementary Fig. 6a as separate channels with remaining images from the same animals. Numbers within images indicate number of RNAs per 100 µm² with standard error of the mean. **c** Left, *iT* hermaphrodites that showed 150–250 generations of continued silencing were crossed with males mutant for RNAi genes (*g(-)*) and resulting descendants homozygous for the mutant allele of the gene were scored at F3 (*rde-1(-)*, *prg-1(-)*, *rrf-1(-)*, *ego-1(-) rrf-1(-)*, *nrde-3(-)*, *hrde-1(-)*), F4 (*ergo-1(-)*, *heri-1(-)*), F5 (*pgl-1(-)*, *rde-3/mut-2(-)*, *mut-16(-)*, *set-32(-)*), ~F15 (*rde-8(-)*) generations, or scored at F4/F6/F8 and pooled (*met-2(-)*). Use of *prg-1(−/+)* parent males owing to the poor mating by *prg-1(-)* parent males is indicated (§). Use of fertile *ego-1(−/+) rrf-1(−/−)* parent hermaphrodites, rather than sterile *ego-1(-) rrf-1(-)* parent hermaphrodites, mated to *iT* males is indicated (#). Use of Cas9-mediated genome editing rather than genetic cross to introduce mutation is indicated (°). Right, Model of piRNA-mediated RNA silencing (see text for details). Scoring of silencing (**a**, **c**) is as in Fig. 1c. Chromosomes with a *dpy* marker (blue font), number (*n*) of animals scored (**c**) or imaged by confocal microscopy (**b**) and scale bar (10 µm) are indicated. Also see Supplementary Figs. 5 and 6. Asterisks indicate $P < 0.05$ using $\chi^2$ test (**c**). Also see 'Genetic Crosses' under Methods. Source data are provided as a Source Data file.

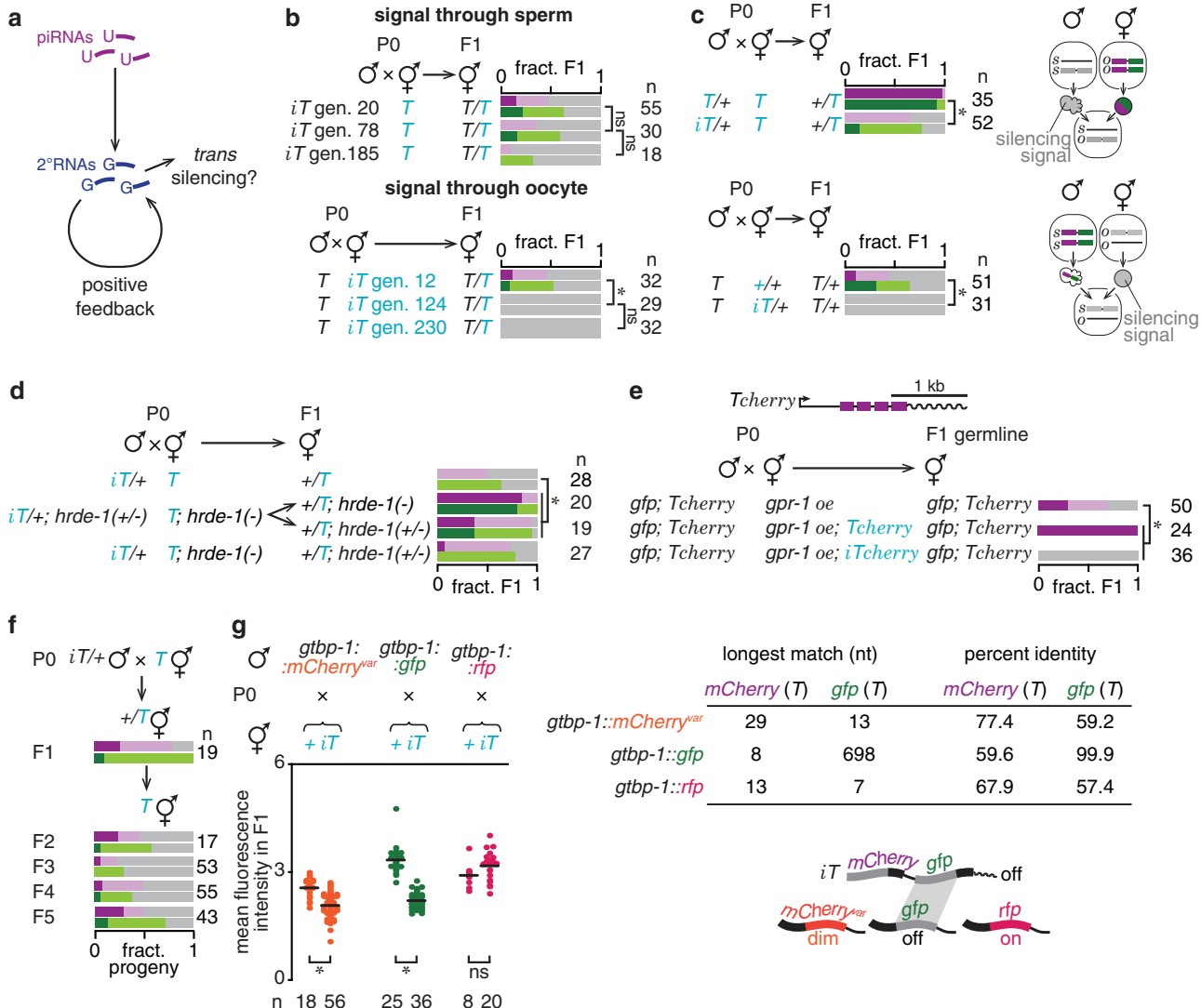

**Fig. 4 Stable silencing after mating is associated with heritable silencing signals that can act in *trans*. a** Schematic showing piRNA-mediated activation of a positive feedback for production of secondary small RNAs (2° RNAs) upon mating-induced silencing. The secondary small RNAs made at one gene could potentially act on other homologous genes (*trans* silencing). **b** Animals expressing *T* were mated with *iT* animals that remained silenced for many generations (*iT* gen. number), and cross progeny were scored. The combined data from each cross is shown in Supplementary Fig. 7a. **c** *T* animals were mated with non-transgenic or hemizygous *iT* animals and cross progeny that inherited only *T* were scored. Schematic: parental presence of *iT* can be sufficient to silence *T* inherited through the other gamete, indicating the inheritance of a separable silencing signal (gray filling). **d** Requirement of *hrde-1* for the activity of the silencing signal was tested by parental, maternal, and/or zygotic loss of HRDE-1. **e** *Tcherry* animals silenced for more than five generations upon initiation of mating-induced silencing were designated as *iTcherry* here. Males expressing *Tcherry* and P*gtbp-1*::*gtbp-1*::*gfp* marker (*gfp*) were mated with *iTcherry* or *Tcherry* hermaphrodites overexpressing *gpr-1* (*gpr-1 oe*). Expression of paternally inherited *Tcherry* in the germline was scored in cross progeny. **f** Silencing of *T* by the separable silencing signal or in *trans* by *iT* was assessed across generations. The remaining results of this cross showing the effect of *trans* silencing are shown in Supplementary Fig. 7c. **g** Males that express homologous (*gfp*) or non-homologous (*mCherry^var*, a synonymous *mCherry* variant or *rfp*) sequences fused to endogenous *gtbp-1* expressed ubiquitously were mated with non-transgenic or *iT* hermaphrodites and fluorescence of GTBP-1::GFP, GTBP-1::mCherry^var or GTBP-1::RFP was quantified in cross progeny (left). Percentage of nucleotide identity and length of the longest continuous match shared by different fluorescent protein coding sequences in pairwise alignment using the Needleman–Wunsch algorithm (penalties for mismatch = 0.1, gap = −1, and gap extension = −0.2) with the *mCherry* or *gfp* sequence of *T* as reference are shown (right top). Schematic summary of homology-dependent *trans* silencing (right bottom). See Supplementary Data files 1–6 for each pairwise sequence alignment. Scoring of silencing (**b**–**f**) is as in Fig. 1c. Chromosomes with a *dpy* marker (blue font) and number of animals scored (*n*) are indicated. Asterisks indicate $P < 0.05$ using $\chi^2$ test (**b**–**d**), Wilson's estimates for proportions (**e**) or two-sided Student's *t*-test (**g**), and ns indicates $P > 0.05$ using the same tests. Also see Supplementary Fig. 7 and 'Genetic Crosses' under Methods. Source data are provided as a Source Data file.

qualitatively distinct (e.g., different timing of small RNA production or different nature of small RNAs). Consistent with *trans* silencing being homology-dependent, *iTΔ* established by mating-induced silencing after deleting *gfp* from *T* did not silence other *gfp* genes in *trans* (Supplementary Fig. 7f). In all cases, silencing (in *trans* or by the separable silencing signal) was restricted to the germline. Furthermore, maternal but not paternal transmission of the silencing signal affected expression of homologous genes (Supplementary Fig. 7g). This difference in efficacy and/or transmission of the silencing signal could reflect differences in the intracellular environment in the two gametes and/or differences in the nature or

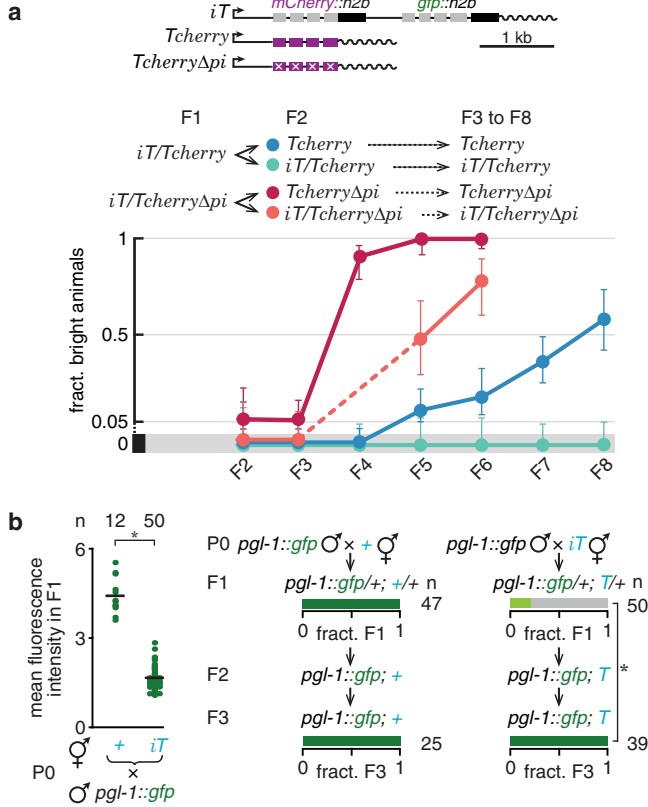

**Fig. 5 *Trans* silencing within the germline can be followed by recovery of gene expression and resistance to subsequent silencing within a few generations. a** *Tcherry* or *TcherryΔpi* animals were mated with *iT* stably silenced for >150 generations and fractions of animals with bright *Tcherry* or *TcherryΔpi* expression were scored in resulting cross progeny (F1) and their descendants (F2 through ≤F8). Error bars indicate 95% confidence intervals. Schematics of transgenes used are indicated above the graphs. See Supplementary Fig. 7j, k for number of animals analyzed in each generation for each genotype and for pedigree. Fractions of bright animals are plotted and error bars indicate 95% CI. **b** *pgl-1::gfp* animals were mated with non-transgenic or *iT* animals and fluorescence of PGL-1::GFP was quantified in cross progeny (left) or scored as in Fig. 1c in cross progeny and their descendants (middle and right). Chromosomes with a recessive *dpy* marker (blue font) and number of animals scored (*n*) are indicated. Also see Supplementary Fig. 7. Asterisks indicate *P* < 0.05 using two-tailed Student's *t*-test (b, left) or $\chi^2$ test (**b**, right). Source data are provided as a Source Data file.

levels of silencing signal inherited through the two gametes[17,46].

**Genes can recover from silencing and become resistant to *trans* silencing.** Sensitivity of *T* to TEI was previously observed when ancestral exposure to neuronal double-stranded RNA (dsRNA) resulted in >25 generations of silencing[47]. To explore whether changes that alter expression of *T* always result in permanent silencing, we used *trans* silencing as an alternative method to initiate silencing and examined the frequency with which recovery of gene expression can occur in the germline. Interaction of *T* with *iT* resulted in strong *trans* silencing of *T* as expected but also weak reactivation of expression from descendants of *iT* (Supplementary Fig. 7h). This reactivation could be mediated by the activity of protective signals opposing silencing signals (Supplementary Fig. 7i), leading to a small fraction of the descendants of *iT* animals showing expression in every generation (Supplementary Fig. 7h). However, the *trans* silencing effect of *iT* on both

*Tcherry* and *TcherryΔpi* (Fig. 5a and Supplementary Fig. 7j, k) was less robust. One generation of exposure to *iT* resulted in silencing even in the absence of *iT* in descendants with homozygous *Tcherry* and *TcherryΔpi*, but expression recovered within a few generations. About 60% of *Tcherry* animals and almost 100% of *TcherryΔpi* animals recovered within seven generations (Fig. 5a). Intriguingly, *TcherryΔpi* became resistant to *trans* silencing despite the continued presence of the silenced *iT* (compare *iT/Tcherry* vs. *iT/TcherryΔpi* in Fig. 5a). The continued silencing of *iT* was indicated by the absence of nuclear-localized GFP and mCherry. These observations suggests that piRNAs binding to a target transcript (*Tcherry*) or perfect homology to *iT* promotes its continued susceptibility to *trans* silencing by small RNAs made from *iT*. Since endogenous genes expressed within the germline are thought to have 'licensing' features that antagonize silencing by piRNAs (e.g., PATC sequences[14], CSR-1 targeting[24]), we examined *trans* silencing of an endogenous gene tagged with *gfp*. The *pgl-1::gfp* gene exhibited a switch from complete *trans* silencing by *iT* in the first generation to undetectable silencing within two generations (Fig. 5b). These results provide two surprising insights into RNAs associated with stable silencing: (1) they are not sufficient for inducing stable silencing at homologous genes even after successful silencing of these genes for a few generations; (2) their activity can be opposed by signals derived from recently active homologous genes despite initial silencing.

**Persistence of silencing by dsRNA depends on the regulatory context of the target gene.** Several studies have reported TEI under diverse conditions, but variations between studies preclude a consistent explanation for susceptibility to TEI (Supplementary Table 1). We therefore simultaneously used identical experimental conditions of feeding RNAi[48] to target identical *gfp* sequences expressed as part of low or single-copy germline genes in parent animals and examined silencing in their untreated descendants (F1–F5). Because parental dsRNA can be deposited into progeny[15,16], only silencing that persists beyond the F2 generation can be unambiguously considered as transgenerational. Out of six target genes tested, two genes showed silencing up to F2 progeny, but only *T* showed silencing beyond F2 (Fig. 6a and Supplementary Fig. 8a–f). Therefore, transgenerational silencing is variable even when targeting the same coding sequence expressed within the same tissue under different regulatory contexts. Even for *T*, while silencing could be maintained upon unbiased propagation, some animals could recover from silencing in later generations (Fig. 6b). Similar to recovery from *trans* silencing (Fig. 5), descendants showed recovery despite silencing in parents (Fig. 6a). The reason for persistent RNA silencing versus recovery from RNA silencing cannot be attributed solely to HRDE-1[40] because silencing was not stable at all target genes despite being HRDE-1-dependent (Supplementary Fig. 8g). We investigated if enhancing silencing by dsRNA could overcome recovery to increase the duration of TEI. Removal of three proteins shown to oppose silencing, the endonuclease ERI-1[49], HERI-1[44], or MET-2[50], enhanced persistence of silencing (Fig. 6c, d and Supplementary Fig. 8h), albeit to a much lesser extent than previously reported for other target genes[44,50]. Thus, *T* and its variants are genes with rare regulatory contexts that enable coding sequences to overcome recovery and retain changes in expression for many generations.

Collectively, we propose that our observations on the response to induced RNA silencing reveal two types of regulatory contexts that drive expression within the *C. elegans* germline (Fig. 7): type I are vulnerable to permanent change in response to transient perturbations and type II are either resistant to perturbations or

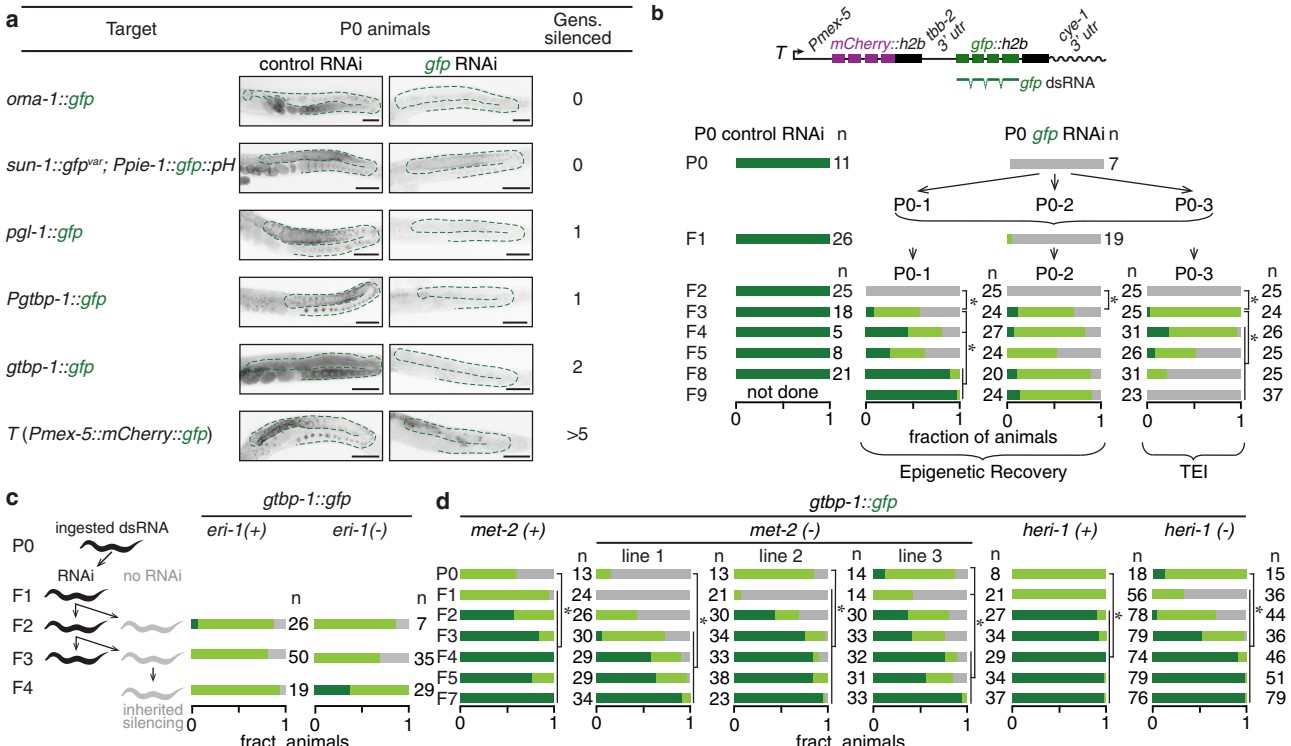

**Fig. 6 Genes silenced by dsRNA in parents commonly recover from silencing in descendants. a** Six target genes containing the same *gfp* sequence were exposed to the same sources of control RNAi or *gfp* RNAi. P0 animals were fed dsRNA for 24 h, and the P0 animals and their untreated descendants for up to five generations (F1–F5) were analysed. Representative images highlight the germline (green outline) of P0 animals. Numbers of descendant generations that show silencing are indicated. Note that *sun-1::gfp^{var}* is also present in the animal with *PH::gfp*, but is not silenced because its *gfp^{var}* sequence only has <14-nt of continuous homology with the *gfp*-dsRNA used for feeding RNAi. Scale bars indicate 50 μm. **b** P0 animals expressing *T* were exposed to control RNAi (P0 control RNAi) or *gfp*-dsRNA (P0 *gfp* RNAi) for 24 h and silencing was analysed in P0 animals and in their untreated descendants. Upon *gfp* RNAi, P0 and F1 animals were each pooled for imaging but subsequent generations each descending from one P0 ancestor (P0-1, P0-2, or P0-3) were imaged as individual isolates. All generations shown were scored by imaging except F2s, which were scored by eye. While one isolate showed TEI, the other two isolates recovered expression (epigenetic recovery). As indicated with a schematic, the sequence of *gfp* dsRNA matches the exons of *gfp* coding sequence. **c** *gtbp-1::gfp* animals were fed *gfp*-dsRNA (black) for one, two or three consecutive generations and their untreated progeny (gray) in a wild-type (*eri-1(+)*) or *eri-1(−)* background were scored for expression of GFP. **d** *gtbp-1::gfp* hermaphrodites in wild-type (*met-2(+)* and *heri-1(+)*), *met-2(−)* or *heri-1(−)* backgrounds were fed *gfp*-dsRNA for 24 h and untreated descendants in subsequent generations (F1–F7) were scored as in Supplementary Fig. 8c. Feeding RNAi of other strains was performed concurrently, thus data for *gtbp-1::gfp* here is the same as in Supplementary Fig. 8c. In *heri-1(-)* animals, the statistical difference between P0 and F1–F2 is due to increased silencing, but that between P0 and F3–F7 is due to decreased silencing. Most animals fed control RNAi and descendants showed bright expression of GFP (except 2 out of 45 F5 descendants and 1 out of 37 F7 descendants of *heri-1(-)* animals that showed dim expression). Number of animals scored (*n*) are indicated. Asterisks indicate $P < 0.05$ using $\chi^2$ test. Also see Supplementary Fig. 8. Source data are provided as a Source Data file.

can recover from them within a few generations. Additional work is needed to discover the particular regulatory molecules and their arrangements that distinguish type I genes from type II genes.

## Discussion

The hallmarks of mating-induced silencing are: (1) silencing is initiated upon inheritance only through the male sperm; (2) once initiated, silencing is stable for many generations; (3) transgenerational silencing is associated with a DNA-independent silencing signal that is made in every generation, can be inherited for one generation, and can silence homologous sequences; and (4) maternal exonic sequences can prevent initiation of silencing. While to our knowledge no other known phenomenon shares all of these hallmarks (Supplementary Table 2), phenomena that share some of these features (elaborated in Supplementary Discussion) can inform future mechanistic studies.

Our analysis of the bicistronic operon *T* and its derivatives suggests that competing maternal signals establish gene expression in progeny. While maternally inherited PRG-1 and piRNAs mediate mating-induced silencing of the paternally inherited copy

of *T* in progeny (Fig. 2), silencing is opposed whenever a maternal protective signal is present (Fig. 2f). This protective signal can act away from the maternal genome, and although its identity is currently unclear, two observations constrain possibilities. One, the maternal presence of part of the *mCherry* coding region from *T* can protect both *mCherry* and *gfp* expression (Fig. 2e), suggesting sequence-dependent recognition of unspliced pre-mRNA or DNA as the target to protect in cross progeny. Two, active *T* continues to be susceptible to mating-induced silencing regardless of protection or escape from silencing in previous generations (Supplementary Fig. 5e, f), suggesting that cross progeny need to inherit the maternal protective signal for consistent gene expression in every generation.

This work reveals that the direction of a genetic cross can strongly influence the phenotype of cross progeny (Fig. 1). Additionally, because not every sibling from a cross has the same phenotype, the choice of the sibling selected for further manipulation can have a profound effect. Subsequent transgenerational persistence of silencing can make phenotype independent of genotype, resulting in erroneous conclusions. Thus, when using

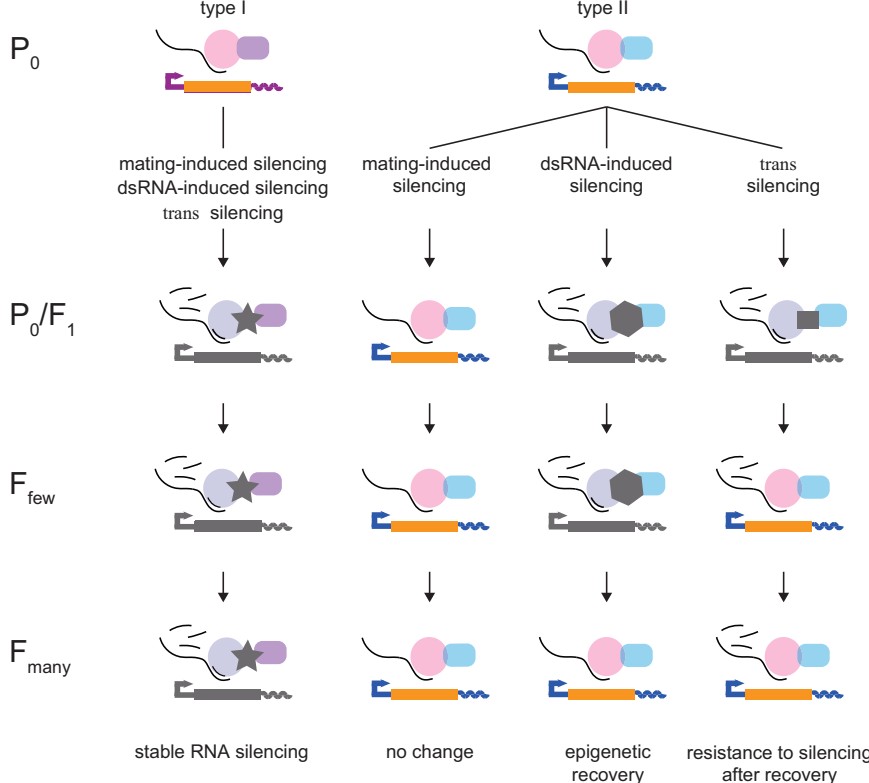

**Fig. 7 Model depicting two types of genes with distinct transgenerational regulation.** Type I genes stably express a recombinant sequence (*T, Tcherry, Tgfp,* etc. described in this study) and yet can undergo permanent heritable change upon RNA silencing initiated using one of multiple methods (left column). Type II genes stably express a recombinant sequence (*gtbp-1::gfp, mCherry::mex-5,* etc. described in this study) and show (1) no change when subject to mating-induced silencing, (2) show silencing for a few generations followed by epigenetic recovery when subjected to dsRNA silencing, (3) show recovery from silencing followed by resistance when subject to *trans* silencing by another silenced gene. We propose that differential recruitment of regulators to the same coding sequence during the P0/F1 generation could explain cases of permanent heritable change versus recovery from change.

genetic crosses to generate strains, both the direction of the genetic cross and choice of the individual cross progeny selected for propagation needs to be controlled for—especially when evaluating epigenetic phenomena. For example, we ensured that every cross was performed with the transgene present in the hermaphrodite to avoid initiating mating-induced silencing in our previous study examining silencing by dsRNA from neurons[47].

The transgenerational stability of mating-induced silencing with potential for recovery of expression even after hundreds of generations (Fig. 3) suggests that this mechanism could be important on an evolutionary time scale. Genes subject to such silencing could survive selection against their expression and yet be expressed in descendants as a result of either environmental changes that alter epigenetic silencing or mutations in the silencing machinery (e.g. in *hrde-1*). This mechanism thus buffers detrimental genes from selective pressures akin to how chaperones buffer defective proteins from selective pressures[51]. Many endogenous genes in *C. elegans* are silenced by HRDE-1[22,40,52,53], some of which could have been acquired when a male with the gene mated with a hermaphrodite without the gene.

There is considerable excitement in the possibility of mechanisms that perpetuate acquired changes and accelerate adaptive evolution[10,54,55]. Our analysis using RNA silencing as an example of induced epigenetic change suggests that the stability of acquired changes is likely to be limited at most genes and that particular regulatory contexts are needed to promote stable epigenetic change. By comparing two different transgenes expressed within the germline, it was proposed that the duration of transgenerational silencing depends on stochastic 'states' adopted by individual organisms[56]. However, examining the same coding sequences with different regulatory contexts (Fig. 6 and Supplementary Fig. 2) suggests that the extent of silencing is not a cell or organism level property, but rather a gene level property. Indeed, different genes within the same tissue can have different genetic requirements for RNA silencing (e.g., *bli-1* silencing but not *dpy-7* silencing requires the nuclear Argonaute NRDE-3[57]). This need for additional regulatory context for the persistence of induced changes is supported by the analysis of RNA-mediated epigenetic changes in yeast[58–60]. Thus, stable epigenetic change requires both a mechanism to copy or amplify induced changes and gene-specific regulatory contexts that recruit or activate this mechanism.

## Methods

**Summary.** All *C. elegans* strains were generated and maintained by using standard methods[61]. Strains were grown at 20 °C, with the exception of some strains with mutations in *prg-1(-)* and *mut-16(-)*, which were grown at 15 °C (see Supplementary Table 3 for full list of strains and Supplementary Table 4 for oligonucleotides used). In all cases, matching control crosses were performed at the same temperature as test crosses. The transgene *T* (*oxSi487*) was introduced into mutant genetic backgrounds through genetic crosses using transgenic hermaphrodites and mutant males to avoid initiation of mating-induced silencing. In all crosses, transgenic genotypes are represented without repetition for simplicity (e.g. '*T*', '*Tcherry*' to refer to homozygous animals *T/T, Tcherry/Tcherry*, respectively). Genotypes represented as '+' are non-transgenic animals with marker mutation(s) (+/+ in colored font) or wild-type animals (+/+ or +). Cross progeny from genetic crosses were identified by balancing or marking *oxSi487* with recessive mutations in *dpy-2(e8) unc-4(e120), unc-4(e120), dpy-2(e8), unc-8(e49) dpy-20 (e1282)* and CRISPR-Cas9 generated alleles of *dpy-10*. In some crosses, cross progeny were identified by genotyping for *oxSi487* transgene using PCR. Genome

editing was performed using Cas9 protein and sgRNAs[62] in most cases (Supplementary Table 5). Silencing of all transgenic strains was measured by imaging under identical non-saturating conditions using a Nikon AZ100 microscope. Quantification of images was performed using NIS Elements (Nikon) and ImageJ (NIH).

**Nomenclature of transgenes.** The letter *T* is used to specify the transgene *oxSi487* in all genetic crosses. The active or expressing allele of *oxSi487* is named as *T* and the inactive or the silenced allele of *oxSi487* is named as *iT* in parents. Genotypes that additionally include a recessive marker (*dpy* or *dpy unc*) are in blue or pink font. See Supplementary Fig. 3 for all variants of *T* and 'Genetic Crosses' for details on recessive mutations used.

**Feeding RNAi and scoring associated defects.** RNAi experiments were performed at 20 °C on nematode growth media plates supplemented with 1 mM IPTG (Omega Bio-Tek) and 25 μg/ml Carbenicillin (MP Biochemicals) (RNAi plates). In all cases genotype- and age-matched animals were fed control RNAi (L4440) and scored alongside as a control.

*Single generation (P0 feeding RNAi).* This assay was performed as described previously[15] and was used in all figures with feeding RNAi except Fig. 6c (see 'Multiple Generations' below). Briefly, L4 animals were fed dsRNA against target genes for 24 h. Some P0 animals were scored for expression while remaining were washed four times in M9 buffer and then allowed to crawl on unseeded plates for an hour to get rid of residual RNAi food. Animals were then singly placed on OP50 and 6–12 L4 animals were blindly passaged every 3–4 days to prevent starvation and to keep track of the generations post feeding. L4 animals were scored in each generation by imaging and L4 siblings were passaged to obtain progeny for the next generation. In feeds performed in Supplementary Fig. 8d and Fig. 6b, F2 animals were scored by eye.

*Multiple generations (P0-F2 feeding RNAi).* Multiple generations of animals (P0-F2) were subjected to feeding RNAi. F1 and F2 animals were scored at L4 stage to assess the potency of the RNAi food and L4 stage siblings were transferred to a new plate with RNAi food to prevent starvation. Similar to the P0 Feeding RNAi protocol, adults (24 h post L4) were washed four times with M9 buffer to remove residual dsRNA and transferred to a plate with OP50. Untreated progeny were then scored for inherited silencing effects. This assay was used in Fig. 6c.

**Expression of dsRNA.** To study inherited silencing, we expressed dsRNA from an extrachromosomal array that is mitotically unstable. Animals that express the array will have both progeny that inherit the array and those that do not. We used an array expressing dsRNA in neurons and DsRed in the pharynx from *jamEx140* [*Prgef-1::gfp-dsRNA::unc-54 3′UTR* and *Pmyo-2::DsRed::unc-54 3′UTR*][47]. Progeny that lack the array were evaluated to measure inherited silencing since parents were exposed to dsRNA from the array but progeny were not. This assay was used in Supplementary Fig. 8h.

**Stages of worms that were imaged using Nikon AZ100 microscope.** Fluorescence intensity of mCherry or GFP was scored after imaging L4-staged animals in all feeding RNAi experiments except in P0 RNAi fed animals and animals expressing *oma-1::gfp* or *Ppie-1::gfp::PH* (Fig. 6a and Supplementary Fig. 8a–f). Fluorescence intensity of mCherry or GFP was scored after imaging L4-staged animals represented in Figs. 1a, c–e, 2a, b, d–g, 3a, c, 4b, c (signal through sperm), d–g, 5, 6 and Supplementary Figs. 1c–k, 2a, b, 3c, 4a, 5a–i, l, n–p, 6d, e, 7a–k, 8a–h (except P0 animals). Fluorescence intensity of mCherry or GFP was scored in adults at 24 h post L4 stage in only P0 animals represented in Fig. 6a, d and in animals represented in Figs. 1e, 2a (*mut-16(-)* animals), g, 4b, c (signal through oocyte), 6b (P0 and F2), Supplementary Figs. 1a, i, 5g, i, m, 6d, 7b, c (F2 through F5), h and i. Fluorescence intensity of mCherry or GFP was scored in adults at 48 h post L4 stage represented in Fig. 4e and Supplementary Fig. 7c (F1s only).

**Genetic crosses.** Three L4 hermaphrodites and 7–13 males were placed on the same plate and allowed to mate in each cross plate. Cross progeny were analyzed 3–5 days after the cross plate was set up. At least two independent matings were set up for each cross. For crosses in Fig. 4f (F1s only), Supplementary Figs. 5a, c (cross with *Mos1/+* only), h, m, n, 7c (F1s only), the required genotypes were determined by PCR (primers P1, P2, and P3) after scoring all animals and only the data from animals with the correct genotypes were plotted. In Figs. 1c–e, 2a, b, g, 3c, 4b–d, 5b and Supplementary Figs. 1c, d, j, k, 2a, 5a, d–g, 6d, e, 7a, b, d–i, *dpy-2(e8)* (~3 cM from *oxSi487*) was used as a linked marker or balancer to determine the genotype of *T*. In Figs. 1e, 2a, f, 3a, 4e, f, 5b and Supplementary Figs. 1f–h, 3d, 5a–d, i, o, p, 7c, j, k, *dpy-10(-)* (~7 cM from *oxSi487*) was used as a linked marker or balancer to determine the genotype of *T*. In Supplementary Figs. 2a, 5d, e, h, *unc-8(e49) dpy-20 (e1282)* was used as a linked marker or balancer to determine the genotype of *ax2053*. In Supplementary Fig. 5a, *unc-4(e120)* (~1.5 cM from *oxSi487*) was used as a linked marker or balancer to determine the genotype of *T*. In Fig. 2a right (control for *rde-1(-)*), *dpy-17(e164) unc-32(e189)* were used as markers to facilitate

identification of cross progeny. Some crosses additionally required identification of cross progeny by genotyping of single worms, including those from Figs. 2b, g, 3c (for *ego-1(-) rrf-1(-)*), 4d, Supplementary Figs. 5a, c (*Mos1/+* only), h, l, 6d, e, 7c, f, g. Animals from crosses with *T; prg-1(+/−)* males in Fig. 2b top or 3c were also genotyped to identify *T/+; prg-1(−/−)* or *T; prg-1(−/−)* cross progeny, respectively. In crosses from Supplementary Figs. 5f, h, 7f, i (control cross), cross progeny of the required genotype was identified by the absence or presence of pharyngeal mCherry or GFP[47], respectively. All strains analyzed for initiation (Fig. 2) and maintenance (Fig. 3) requirements had been mutant for at least two generations, except when testing the requirement for *prg-1(-)* in initiation, which was done using *prg-1(-)* animals that were mutant for one generation or *ego-1(-)* in maintenance, which was done using *ego-1(+/−)* parents.

*Genetic crosses with mut-16 mutants to test for initiation of mating-induced silencing.* In Fig. 2a, L4 male cross progeny were scored for only mCherry fluorescence because GFP fluorescence was difficult to assess in the single gonad arm of the L4 male germline due to gut autofluorescence.

*Genetic crosses to determine if recovery of expression upon removal of wild-type hrde-1 is lost upon re-introduction of wild-type hrde-1.* In Supplementary Fig. 6e, *hrde-1(-)* mutant males were mated with *iT* hermaphrodites that remained silenced for ~270 generations, resulting in cross progeny (F1) that were allowed to produce self-progeny (F2) from which animals homozygous for *T* and for the wild-type or the mutant allele of *hrde-1* were assessed across generations by passaging self-progeny (F3 through F7). In addition, every generation of *hrde-1(-); T* hermaphrodites produced by self-fertilization (F2 through F6) was mated with either wild-type (+/+) or *hrde-1(-)* males to examine the possibility of re-initiation of transgenerational silencing. mCherry and GFP fluorescence was scored in heterozygous F1 cross progeny (*hrde-1(−/+)*) and in F3 or later descendants of genotypes depicted. Cross progeny (gray text) of F2 *hrde-1(-); T* hermaphrodites mated with wild-type males were not obtained despite multiple biological repeats due to experimental design. Specifically, the mating was set up in replicates between a single *hrde-1(-); T* hermaphrodite with three wild-type males at every generation, beginning from the F2 generation onwards. The selection of hermaphrodites of *hrde-1(-); T* genotype was successful only from F3 generation, because homozygous *hrde-1(-); T* could only be set up from the F2 generation, which is the very first generation the genotype of descendants can become *hrde-1 (-); T* after the cross set up at P0. As a result, F2 *hrde-1(-); T* hermaphrodites were needed for crosses, but they could not be distinguished from their *hrde-1(+); T* or *hrde-1(+/−); T* siblings on the F1 > F2 plate. The only way to determine the genotype of the hermaphrodite used was by first mating a single random hermaphrodite of unknown *hrde-1* genotype with three wild-type males, and then allowing for the F3 progeny to be laid for 3 days before sacrificing the F2 hermaphrodite for genotyping. However, by this point, the F2 hermaphrodite would be harboring wild-type sperm in its spermatheca, which could potentially confound the genotyping PCR.

*Genetic crosses using animals overexpressing gpr-1.* To analyze DNA-independent signals we used a recently developed tool that prevents paternal and maternal pronuclei from fusing within the zygote[19,20]. A G protein regulator, GPR-1, when overexpressed maternally, increases forces that pull on spindle poles and prevents the maternal and paternal nuclei from fusing. This allows the contents of the paternal nucleus to be inherited only into cells of the P lineage and the contents of the maternal nucleus to be inherited only into the AB lineage. By way of such non-Mendelian segregation in most cross progeny, paternal DNA is inherited into all germline cells and select somatic cells (such as the intestine and body-wall muscles) and maternal DNA is only inherited into the somatic cells (Fig. 2d). A smaller fraction of progeny either have maternal DNA in the germline and some soma, and paternal DNA in most somatic cells (Fig. 2d), or undergo Mendelian segregation with paternal and maternal DNA in all cells (data not shown). To analyze the robustness of this tool in our hands, we tested the segregation of paternal and maternal DNA using *gtbp-1::gfp*, which expressed cytoplasmic GFP in all tissues (Fig. 2d). When hermaphrodites overexpressing *gpr-1* (*gpr-1 oe*) were crossed with males carrying *gtbp-1::gfp*, >95% of cross progeny showed non-Mendelian segregation with paternal DNA inherited into cells of the P lineage (based on presence of GFP in the germline) and maternal DNA into cells of the AB lineage (based on absence of GFP in some pharyngeal cells and neurons). A much smaller population of cross progeny (<5%) showed either the inverse pattern of segregation or Mendelian segregation. We used *gtbp-1::gfp* as the marker to identify non-Mendelian cross progeny in further crosses with *gpr-1 oe*. To analyze effects of parental signals on *T* in the germline, we had to ensure that *T* (and the accompanying marker gene, *gtbp-1::gfp*) was always inherited from the male because the majority of non-Mendelian cross progeny would inherit paternal DNA into the germline. Since the transgene expressing *gpr-1* also expressed a synonymous variant of *gfp*, we used a variant of *T* i.e., either *TΔΔΔ* or *Tcherry* for further analyses to avoid GFP fluorescence from two different sources, confounding interpretation.

*Genetic crosses with Pmex-5::Tcherry::mex-5 3′utr and Pmex-5::Tcherry::tbb-2 3′ utr.* Integration of *Pmex-5::Tcherry::mex-5 3′utr* and *Pmex-5::Tcherry::tbb-2 3′utr* by MosSCI into the genome resulted in spontaneous silencing of the transgenes[22],

whose expression could be revived by mutation of *hrde-1*. Because parental *hrde-1* was dispensable and zygotic *hrde-1* was sufficient for initiation of mating-induced silencing (Fig. 2b), we used *Pmex-5::Tcherry::mex-5 3′utr; hrde-1(-)* or *Pmex-5::Tcherry::tbb-2 3′utr; hrde-1(-)* parent animals in reciprocal crosses to test for mating-induced silencing (Supplementary Fig. 1i). These crosses resulted in cross progeny of genotypes *Pmex-5::Tcherry::mex-5 3′utr; hrde-1(+/−)* or *Pmex-5::Tcherry::tbb-2 3′utr; hrde-1(+/−)*, respectively, which were then scored for mating-induced silencing.

### Generation and maintenance of *iT* and *iTΔ* strains.

To make hermaphrodites with *iT* linked to a *dpy* marker, AMJ581 hermaphrodites were mated with N2 males to generate cross progeny males that all show bright mCherry fluorescence from *oxSi487*. These males were then mated with N2 hermaphrodites to give cross progeny (F1) with undetectable mCherry fluorescence. F1 animals were allowed to give progeny (F2) that were homozygous for *oxSi487* as determined by the homozygosity of a linked *dpy-2(e8)* mutation. One such F2 animal was isolated to be propagated as the *iT* strain (AMJ692).

To make males with *iT*, *dpy-17(e164) unc-32(e189)* hermaphrodites were mated with EG6787 males to generate cross progeny (F1) hermaphrodites with undetectable mCherry fluorescence. These cross progeny were allowed to have self-progeny (F2) that were homozygous for *oxSi487*. Two such F2s were isolated to be propagated as two different *iT* lines. One of these was designated as AMJ724 and used for further experiments. These strains maintained the silencing of *oxSi487* and were heat-shocked to produce males. Genotypes of *iT* strains were verified using PCR.

To make hermaphrodites with *iTΔ* linked to a *dpy* marker, AMJ767 hermaphrodites were mated with N2 males to generate cross progeny males with bright mCherry fluorescence. These males were then mated with GE1708 hermaphrodites to give cross progeny (F1) with undetectable mCherry fluorescence. F1 animals were allowed to give descendants that are homozygous for *TΔ* as determined by genotyping for *jamSi20*. A homozygous descendant was isolated to be propagated as the *iTΔ* strain (AMJ917). Genotypes of *iTΔ* strains were verified using PCR.

AMJ692 was used to test for recovery of gene expression ~150 generations after it was made. This generation time was estimated as follows: worms were passaged every 3.5 days for 143 generations over a period of 556 days, except for three intervals when they were allowed to starve and larvae were recovered after starvation. These intervals with recovery from starvation spanned a total of ~6 generations over 49 days. Thus, the total number of generations = 143 + ~6 = ~150 generations. The generation times for other *iT* strains, AMJ724, AMJ552, and AMJ844, were similarly estimated. *iT* strain silenced for >150 generations was used to test the requirements for RNAi factors in the maintenance of transgenerational silencing.

### CRISPR-Cas9 mediated editing of *oxSi487*.

To generate edits in *oxSi487*, Cas9-based genome editing with a co-conversion strategy[62] was used. Guide RNAs were amplified from pYC13 using primers listed in Supplementary Table 5. The amplified guides were purified (PCR Purification Kit, Qiagen) and tested in vitro for cutting efficiency (Cas9, New England Biolabs catalog no. M0386S). For most edits, homology template for repair (repair template) was made from gDNA using Phusion High Fidelity polymerase (New England Biolabs catalog no. M0530S) and gene-specific primers to separately amplify regions precisely upstream and downstream of the site to be edited. The two PCR products were used as templates to generate the entire repair template using Phusion High Fidelity Polymerase and the fused product was purified using NucleoSpin Gel and PCR Clean-up (Macherey-Nagel, catalog no. 740609.250). Homology templates to generate *TΔΔ* and *dpy-10(-)* were single-stranded DNA oligos. Wild-type animals were injected with 0.12–12.9 pmol/μl of guide RNAs, 0.08–1.53 pmol/μl of homology repair template to make edits in *T* and in *dpy-10* and 1.6 pmol/μl of Cas9 protein (PNA Bio catalog no. CP01). In animals with *TΔΔ* edit, *Punc-119* deletion resulted in Unc animals due to the *unc-119(ed3)* mutation in the background of EG6787, suggesting that a functional transcript was not made from the remaining part of the rescuing *Punc-119::unc-119::unc-119 3′utr* insertion at *ttTi5605*. Edits were verified using PCR and Sanger sequencing. For additional details on specific reagents, see Supplementary Table 5.

### CRISPR-Cas9 mediated insertion.

To generate large insertions, the Cas9-based editing protocol was adapted from ref. [63]. The following mix was injected into HT1593 animals: 42–55 ng/μl plasmid expressing Cas9 protein and sgRNA sequence specific to chromosome II site near *ttTi5605* (pDD122) or chromosome I site near *ttTi4348* (pSD18), 105 ng/μl of pMA122 (*Phsp-16.41::peel-1::tbb-2utr*), 42–55 ng/μl of repair plasmid for insertion of *Tcherry^Crispr^* (*jamSi38*, *jamSi40*, *jamSi41*) or *Tcherry* I (*jamSi56*). Following injection, animals were singled out and the plate was allowed to crowd until starvation. Starved plates were heat-shocked at 34 °C for 2.5–4 h and heat-shocked animals were allowed to recover overnight. Non-Unc animals that survived the heat shock were singled out, propagated and screened for the edit using PCR. Single-copy insertions were then verified in isolates that screened positive for the edit after extraction of genomic DNA.

### Mos-mediated single-copy insertion (MosSCI).

To generate large insertions, the MosSCI protocol was adapted from ref. [11]. The following mix was injected into EG4322 animals: 50–55 ng/μl plasmid expressing Mos1 transposase (pCFJ601: *Peft-3::mos1 transposase::tbb-2utr*), 105 ng/μl of pMA122 (*Phsp-16.41::peel-1::tbb-2utr*), 50–55 ng/μl of repair plasmid for insertion of *Tcherry*, *Tgfp*, *TcherryΔpi*, *Tcherry::tbb-2 3′ utr* or *Tcherry::mex-5 3′ utr* into chromosome II near *ttTi5605* insertion site. Following injection, animals were singled out and the plate was allowed to crowd until starvation. Starved plates were heat-shocked at 34 °C for 2.5–4 h and heat-shocked animals were allowed to recover overnight. Non-Unc animals that survived the heat shock were singled out, propagated and screened for the edit using PCR. Single-copy insertions were then verified in isolates that screened positive for the edit after extraction of genomic DNA.

### Quantitative RT-PCR (RT-qPCR).

Total RNA was isolated using TRIzol (Fisher Scientific) from 50 to 100 μl pellets of mixed-stage animals. Three biological replicates were isolated by pelleting animals from three different plates of the same strain. RNA was extracted by chloroform extraction, precipitated using iso-propanol, washed with ethanol and resuspended in 20–30 μl of nuclease-free water. 2–5 μl of resuspended RNA was set aside to run on a gel and the remaining was DNase-treated in DNase buffer (100 mM Tris-HCl, pH 8.5, 5 mM CaCl$_2$, 25 mM MgCl$_2$), and incubated with 0.25 μl DNase I (New England Biolabs, 2 units/μl) at 37 °C for 60 min followed by heat inactivation at 75 °C for 10 min. Pre- and post-DNase treated RNA were run on a 1% agarose gel to check for the presence of rRNA bands. RNA concentration was measured and equal amounts (500–5000 ng) of RNA were converted to cDNA using SuperScript III Reverse Transcriptase (Invitrogen catalog no. 18080044) with two-fold reduced quantities compared to manufacturer's recommendations. For cDNA conversion, 3–5 technical replicates were done for each biological replicate of each sample and RT primer P82 was used for *R11A8.1*, P176 for *tbb-2* pre-mRNA, P177 *for tbb-2* mRNA, P83 for *mCherry* pre-mRNA P84 for *mCherry* mRNA, P78 for *gfp* pre-mRNA and P85 for *gfp* mRNA. PCR was performed with the cDNA as template and using LightCycler 480 SYBR Green I Mastermix (Roche catalog no. 4707516001) guidelines according to the manufacturer's recommendations. For analysis of pre-mRNA, primers P86 and P87 were used for *R11A8.1*, P88 and P89 were used for *tbb-2*, P93 and P179 were used for *mCherry*, and P96 and P97 were used for *gfp*. For analysis of mRNA, primers P94 and P95 were used for *tbb-2*, P90 and P91 were used for *mCherry*, and P98 and P99 were used for *gfp*. Fold change was calculated using $2^{-Ct}$ values and samples were normalized to total RNA.

Three (Supplementary Fig. 2c) to six (Supplementary Fig. 6c) independent biological replicates were typically measured, with each biological replicate being the median of three to five technical replicates. A scaled scatter plot was used to depict the relative abundance of pre-mRNA and mRNA for each biological replicate. RNA abundance was estimated as proportional to $2^{-Ct}$ and target transcripts were normalized to total RNA to obtain relative abundance.

### Chromatin immunoprecipitation-qPCR (ChIP-qPCR).

This protocol was adapted from ref. [64] 300–500 μl of frozen mixed-stage worm pellets were used for each ChIP experiment. Three biological replicates were done for every strain and worms from each sample were split into 100 μl pellets. Frozen pellets were crushed by grinding with a mortar and pestle. Crushed pellets were resuspended in 1 ml buffer A (15 mM HEPES-Na, pH 7.5, 60 mM KCl, 15 mM NaCl, 0.15 mM beta-mercaptoethanol (CALBIOCHEM catalog no. 444203), 0.15 mM spermine (Sigma-Aldrich catalog no. S3256-1G), 0.15 mM spermidine (Sigma-Aldrich catalog no. S2626-1G), 0.34 M sucrose, 1X HALT protease (ThermoScientific catalog no. 78440) and phosphatase inhibitor cocktail (ThermoScientific catalog no. 78440)). To crosslink, formaldehyde was added to a final concentration of 2%, and incubated at room temperature for 15 min. The formaldehyde was quenched by adding 0.1 ml 1 M Tris HCl (pH 8.0). The lysate was spun at $15,000 \times g$ for 1 min at 4 °C. The resulting pellets were washed twice with ice-cold buffer A by centrifuging between washes. The pellets were resuspended in 0.3 ml buffer A with 2 mM CaCl$_2$. Micrococcal nuclease (Roche catalog no. M0247S) was added to a final concentration of 0.3 U/μl and incubated for 5 min at 37 °C (the tubes were inverted several times per minute). EGTA to a final concentration of 20 mM was added to stop the digestion reaction and samples were centrifuged at $15,000 \times g$ for 1 min at 4 °C, followed by washing the resulting pellets with 300 μl of ice-cold RIPA buffer (1X PBS, 1% NP40 (Spectrum catalog no. T1279), 0.5% sodium deoxycholate (Sigma-Aldrich catalog no. D6750-10G), 0.1% SDS, 1X HALT protease and phosphatase inhibitor and 2 mM EGTA (Sigma-Aldrich catalog no. E3889-10G)). Samples were centrifuged at $15,000 \times g$ for 1 min at 4 °C. The pellet was resuspended after washes in 0.8 ml ice-cold RIPA buffer, and solubilized by shearing using the Covaris[65]. Samples were kept on ice at all times except during shearing. All sheared lysates for each biological replicate were pooled and split equally to precipitate for all chromatin marks being measured. Sheared lysates were centrifuged at $15,000 \times g$ for 2 min. 80 μl of the supernatant was set aside at −20 °C for "input" libraries and the remaining supernatant was used for IP. Antibodies were chosen based on their efficiency in *C. elegans*[66]. One of 2 μg of anti-H3 antibody (Abcam, ab1791), 3 μg of anti-H3K9me1 antibody (Abcam, ab8896), 3 μg of anti-H3K9me2 antibody (Abcam, ab1220) or 2 μg of anti-H3K9me3 antibody (Abcam, ab8898) was added and agitated gently at 4 °C overnight. 50 μl of protein A Dynabeads (10% slurry in 1x PBS buffer) was added and mixed by shaking for 2 h

at 4 °C. The beads were then washed four times (4 min/wash) with ice-cold 600 µl LiCl washing buffer (100 mM Tris HCl, pH 8, 500 mM LiCl, 1% NP-40, 1% Sodium deoxycholate). A magnetic stand (DynaMag-2 Magnet, Thermo Scientific) was used to pellet beads and the supernatant was discarded after every wash. Beads and input were incubated with 450 µl worm lysis buffer (0.1 M Tris HCl, pH 8, 100 mM NaCl, 1% SDS) containing 200 µg/ml proteinase K at 65 °C for 4 h with agitation every 30 min to elute the immunoprecipitated nucleosome and reverse crosslinks. DNA was isolated by organic extraction and precipitation. DNA obtained was measured by qPCR using LightCycler 480 SYBR Green I Mastermix according to the manufacturer's recommendations (see PCR portion of qRT-PCR method for details and pre-mRNA, or equivalently DNA, primers for *R11A8.1, mCherry* and *gfp*). Fold change was calculated using $2^{-\Delta\Delta Ct}$ method and samples were normalized to co-immunoprecipitated control gene, *R11A8.1*.

**Single-molecule fluorescence in situ hybridization (smFISH)**. Custom Stellaris FISH probes were designed against only exons of *mCherry* and *gfp* sequence from *oxSi487* using the web-based Stellaris FISH Probe Designer from Biosearch Technologies (www.biosearchtech.com/stellarisdesigner). Any probe design expected to span exon-exon junctions was avoided to allow for the equivalent detection of both mature and nascent transcripts. Standard *C. elegans* smFISH protocol followed by 4′,6-diamidino-2-phenylindole (DAPI) staining was used as described[67,68]. The probe blend to detect *mCherry* includes 25 exon-specific probes (P112 through P136) each tagged with Quasar 670 dye and antisense to *mCherry* RNA. The probe blend to detect *gfp* includes 26 exon-specific probes (P137 through P162) each tagged with Quasar 670 dye and antisense to *gfp* RNA. The adapted smFISH protocol is as follows: 50–100 L4 animals or adult animals ~24 h post L4 (Fig. 3b, Supplementary Figs. 2e, 4b, 6a) were paralyzed in 400 µl 1X Phosphate Buffered Saline 0.1% Tween-20 (PBST, Amresco, catalog number C999G23 K875-500ML) containing 0.25 mM levamisole for dissection or whole animals younger than L4 (Supplementary Fig. 4b) were washed in 1X PBST and fixed in 1 ml fix solution (3.7% formaldehyde (Amresco, catalog number 0493-500 ML) in 1X PBST) on a nutator at room temperature. Fixation time ranged between 15 and 45 min across different trials. Samples were washed in 1X PBST, incubated for 10 min in permeabilizing solution (0.1% Triton X-100 in 1 ml of 1X Gibco PBS pH 7.4 (Thermofisher Scientific, catalog number 10010023)), washed twice in PBST and resuspended in 1 ml 70% ethanol and incubated between 1 and 7 days at 4 °C. Fixed animals were then equilibrated and washed with wash buffer (2X Sodium Saline Citrate (SSC, Sigma Aldrich, catalog number 11666681001), 10% formamide (Millipore Sigma, catalog number 4650-500 ML or Amresco, catalog number 0314-500 ML), 0.01% Tween-20 (Fisher Scientific, catalog number BP337-100)) hybridized with 0.025 µM probes diluted in hybridization buffer (10% dextran sulfate (Sigma Aldrich, catalog number D8906-5G), 2X SSC, 10% formamide) for 48 h in a 37 °C rotator in the dark. Hybridized animals were then washed in wash buffer, incubated with DAPI solution (1 µg/ml DAPI in wash buffer) for 30–120 min protected from light, washed twice in wash buffer for 5 min each in a rotator and used for mounting. Worms were resuspended and incubated for 5 min at room temperature or up to 6 h at 4 °C in a GLOX buffer without enzymes (2X SSC, 1% glucose (Fisher Scientific, catalog number D16-500), 0.1 M Tris pH 8.0 (Thermofisher Scientific, catalog number AM9855G) in RNase-free water), treated with freshly made GLOX-enzyme buffer (100 µl GLOX buffer, 1 µl glucose oxidase (MP Biomedicals/Fisher Scientific, catalog number 0219519610), 3.7 mg/ml, 1 µl catalase (Fisher Scientific, catalog number S25239A), 1 µl 200 mM Trolox (Acros Organics/Fisher Scientific, catalog number 218940050)) and prepared for imaging by dropping the sample on a coverslip followed by placing and sealing on a microscope slide with a mix of Vaseline, lanoline, and paraffin. All samples within a single experimental set included control strains and were subjected to identical conditions (e.g. incubation times) to minimize variability within the experiment. RNase-free conditions were used in all smFISH experiments.

AMJ1259, AMJ1260, and AMJ1261 females were mated with AMJ1045 or EG6787 males and extruded gonads of cross progeny hermaphrodites staged at ~24 h post L4 were subjected to smFISH protocol using *mCherry* probes (Supplementary Fig. 4b). For Supplementary Fig. 6a, b, extruded gonads of EG6787 ("*T*"), AMJ552 ("*iT*"), and N2 ("wild type") adult hermaphrodites staged at ~24 h post L4 were subjected to the smFISH protocol using either *mCherry* or *gfp* probes. For Supplementary Fig. 2e top row, extruded gonads of EG6787, AMJ1170, JH3323, and N2 adult hermaphrodites staged at ~24 h post L4 were subjected to the smFISH protocol using *mCherry* probes alone. For Supplementary Fig. 2e bottom row, extruded gonads of EG6787, AMJ1195, JH3197, and N2 adult hermaphrodites staged at ~24 h post L4 were subjected to the smFISH protocol using *gfp* probes alone.

**Confocal microscopy to image single-molecule RNA signals or protein fluorescence**. Images were taken using Leica SP5 confocal microscope with the ×63 oil immersion objective at 500% digital zoom for smFISH samples and 400% digital zoom to capture protein fluorescence. A single confocal slice of 0.5 µm thickness was captured at regions corresponding to distal, loop, or proximal regions of the dissected gonad. The Z position was oriented to be the same plane as the nucleus of the distal tip cell for all three regions imaged in most dissected gonads. To image whole worms between L2 and L3 stages for smFISH, a Z stack of a part of the germline that could be accommodated within the field of view at the same

magnification as was used for dissected gonads was imaged with a step size of 0.5–1 µm and displayed as a maximum intensity projection. Brightfield and DAPI images were taken using photomultiplier tubes whereas *mCherry* and *gfp* RNA and protein fluorescence images were taken using Hybrid Detector (HyD). For both smFISH and protein fluorescence, the XY laser scan was set to 400 Hz and imaged at a resolution of 1024 × 1024 pixels. Quasar 670 probes were excited using Alexa 633 nm laser (50% White Light Laser) and signal was detected between 650 and 715 nm with the pinhole at 105.05 µm. DAPI was excited using 405 nm (3–30% UV laser) and signal was acquired between 422 and 481 nm with the pinhole at 95.52 µm. For Quasar 670 and mCherry or GFP protein fluorescence, a line average of 6–8 with 1–2 frame accumulation was used. For DAPI, 3–4 line average was used.

**Quantification of silencing and measurement of fluorescence intensity**. To classify fluorescence intensity after imaging, animals of the L4 stage or 24 h after the L4 stage were mounted on a slide after paralyzing the worm using 3 mM levamisole (Sigma-Aldrich, Cat# 196142), imaged under non-saturating conditions (Nikon AZ100 microscope and Photometrics Cool SNAP HQ[2] camera), and binned into three groups—bright, dim and off. A C-HGFI Intensilight Hg Illuminator was used to excite GFP or Dendra2 (filter cube: 450–490 nm excitation, 495 dichroic, and 500–550 nm emission), or mCherry or RFP (filter cube: 530–560 nm excitation, 570 dichroic, and 590–650 nm emission). Sections of the gonad that are not obscured by autofluorescence from the intestine were examined to classify GFP and mCherry fluorescence from *oxSi487*. Autofluorescence was appreciable when imaging GFP or Dendra2 but not when imaging mCherry.

In some cases, fluorescence intensity within the germline was scored by eye without imaging at L4 stage in Fig. 6c and Supplementary Fig. 8h or at 24 h after the L4 stage in Supplementary Figs. 5e, f (to find bright male F1s), 7b (to find off male F1s) and 8a at fixed magnification and zoom using the Olympus MVX10 fluorescent microscope without imaging.

To quantitatively measure protein fluorescence of mCherry and GFP from *T* imaged using Nikon AZ100 as described above (Fig. 1c) and protein fluorescence from other transgenes (Figs. 4g, 5b left and Supplementary Fig. 8a–f), regions of interest (ROI) were marked using either NIS elements or ImageJ (NIH) and the intensity was measured. Background was subtracted from the measured intensity for each image. For Figs. 1c, 4g, 5b and Supplementary Fig. 8a–f, fluorescence intensity was measured as x − b, where x = mean intensity of ROI and b = mean intensity of background. The obtained intensity values were converted to a $\log_2$ scale and plotted.

In experiments with feeding RNAi, target gene (*gfp*) and control RNAi fed animals for each strain were imaged at the same exposure. Control and experimental animals were all imaged at non-saturating conditions either at a fixed exposure or by setting exposure to their respective controls. Previous reports have suggested that the pharynx, neurons, and vulval muscles can be resistant to silencing by dsRNA[69,70] and hence were not included in our scoring.

All images being compared were adjusted identically using Adobe Photoshop for display.

*Quantification of expression from Tgfp.* Insertion of *Tgfp* into the genome resulted in variable GFP expression in all animals. However, in the case of mating-induced silencing, silenced animals displayed no detectable silencing of GFP as measured by quantification. To quantitatively measure fluorescence of GFP from *Tgfp* (Supplementary Fig. 1f), ROI of the germline that excluded the intestine was marked using Fiji (NIH) and the intensity was measured. An area outside the worm within the same image was measured for background intensity. The mean fluorescence intensity from *Tgfp* expression was calculated by subtracting the background intensity from measured GFP intensity as described above.

**Quantification of smFISH signals**. Leica images (.lif format) were opened in Fiji (NIH), display range was adjusted, background was subtracted twice sequentially using a rolling ball radius of 50 pixels (~2.7 µm), threshold was adjusted, and number of RNA dots ≤250 object voxels in size were quantified per unit area. All parameters were adjusted identically among images of strains being compared. All images being compared were adjusted identically using Adobe Photoshop for display.

Colocalization between smFISH signals of *mCherry* and *gfp* transcripts (Fig. 1b and Supplementary Fig. 1b) was done using Colocalization Colormap plugin within Fiji. The plugin finds the correlation between pixel intensities from the same spatial coordinates between two images using the following formula: [(intensity of the pixel in *mCherry* RNA image − average pixel intensity in *mCherry* RNA image) × (intensity of the pixel in *gfp* RNA image − average pixel intensity in *gfp* RNA image)]/[(maximum pixel intensity in *mCherry* RNA image − average pixel intensity in *mCherry* RNA image) × (maximum pixel intensity in *gfp* RNA image − average pixel intensity in *gfp* RNA image)]. The values of the correlation will range between 1(most colocalized) to −1(least colocalized) and is represented by a scale in the figure. The original mCherry images contained fluorescence signal from both mCherry::H2B (which represents nuclear chromatin) and from *mCherry* RNA probed with Quasar 570 probes. These images captured both mCherry protein and RNA signals because of a large overlap between fluorescence spectra from mCherry and Quasar 570. Using these images, we obtained an approximation of *mCherry* RNA signal by subtracting the mCherry::H2B signal using DAPI images whose

signals coincided with mCherry::H2B signals, but not with *mCherry* RNA signals. The resulting subtracted image (mCherry protein + *mCherry* RNA − DAPI) was then used to examine the extent of colocalization between *mCherry* RNA and *gfp* RNA (Fig. 1b and Supplementary Fig. 1b).

**Statistical analyses, reproducibility, and plotting**. For each figure, $\chi^2$ test was used to compare data as indicated in figure legends except in cases where only one category (bright or silenced) was present in both datasets being compared, in which case Wilson's estimates for proportions was used. All comparisons shown include comparisons between only GFP fluorescence or only mCherry fluorescence within each experiment. Significance for ChIP and qRT-PCR experiments and crosses were compared using Student's *t*-test (Fig. 2a (*Tgfp*), Supplementary Figs. 1f, 2c, 6c, f, g). Matlab, R, and Microsoft Excel were used to plot fluorescence intensity (bar chart, rose plot, box and whisker plot, dot plot, line plot) and qPCR data. Exact *P* values are provided in Source Data. The biological replicates for each experiment are as indicated in figures, figure legends, and methods. Experiments were performed once or the number of times indicated within Source Data as independent cross plates for genetic crosses. Critical experiments were replicated by multiple authors (mating-induced silencing by 8 authors, *pgl-1*-dependent silencing by two authors). In these cases, results were reproducible. Representative images presented were from different imaging sessions, where similar patterns were observed with number of imaging sessions as indicated here: Fig. 1a shows representative images from >10 sessions of >10 animals; Fig. 1c shows representative images from 1 session of 5–11 animals of each category; Fig. 2c mosaic expression from 2 sessions and 92 animals (mosaicism was also tested with *sur-5::gfp*, which showed similar results); Fig. 6a representative RNAi from 1 session and 7 animals; Fig. 1b and Supplementary Fig. 1b representative figure of colocalization from 2 sessions and 6 gonads.

**Reporting summary**. Further information on research design is available in the Nature Research Reporting Summary linked to this article.

## Data availability
All data generated or analyzed during this study are included in this published article (and its Supplementary Information files). More than 10,000 images were generated during this study to document expression levels. These images and other data supporting the finding of this study are available from the authors upon reasonable request. Source data are provided for this paper at https://doi.org/10.6084/m9.figshare.14642820.

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

## Acknowledgements

We thank Nathan Shugarts for most of the Sanger sequencing of *oxSi487*, referred to as *T* within the manuscript, presented in Supplementary Fig. 1a; members of the Jose laboratory, Norma Andrews, Karen Carleton, Steve Mount, and anonymous reviewers for comments on the manuscript; the *Caenorhabditis elegans* Genetic Stock Center, the Seydoux laboratory (Johns Hopkins University), the Cohen-Fix laboratory (National Institutes of Health), the Fire laboratory (Stanford University), the Bringmann laboratory (Max Planck Institute) and the Hunter laboratory (Harvard University) for worm strains. This work was supported in part by National Institutes of Health Grants R01GM111457 and R01GM124356 to A.M.J.

## Author contributions

All authors contributed to experimental design and analysis. S.D., P.R., S.A., F.E., M.D., Y.L, Y.E.C, and M.C. performed experiments. S.D., P.R., M.C., and A.M.J. wrote the manuscript. All authors edited the manuscript.

## Competing interests

The authors declare no competing interests.
