## [Peer Review File · Nature Communications]

REVIEWER COMMENTS

Reviewer #2 (Remarks to the Author):

This much revised version of the Devanapally and Raman et al paper has improved greatly and focuses on mating induced silencing mechanism that the authors have found in *C. elegans*. The central finding is that mating can trigger silencing of non-self sequences via a piRNA mediated pathway. This silencing is remarkably stable >300 generations as opposed to other types of silencing mechanisms that only last for a few generations (~5-6). The paper is robust and thoughtful, and it has very impressive genetics (quality and quantity). In its current form, the concepts are easy to understand, the paper flows well, has a central topic and will definitely be an advance to the field. Although the *C. elegans* field is not a stranger to dsRNA silencing mechanisms and their transgenerational inheritance, this is a unique case where the initiation and maintenance of (and protection from) silencing are all derived from RNAs but in distinct pathways. This is enumerated in the paper through careful experimentation and reasoning. It highlights the versatility of RNA silencing mechanisms that can be used by an organism and has uncovered a reproducible and extremely strong method of transgenerational silencing that the authors hope will be used widely by the community for future studies. I strongly support the publication of this paper with a few changes made to the text, as follows.

Major comment:

1. In the interest of highlighting the maternal contribution in the paper, it would serve the authors well to discuss that maternal transcripts provide both the silencing and the protection by pointing out the various lines of evidence that support that view. In the reasoning for figure 2: the authors point out that either oocyte licensing explains it all or that T (inherited through the sperm) remains protected in perpetuity. The key finding for this is that a protected cross progeny is still susceptible to silencing from mating with a mother that has never seen this sequence (Figs S5e and f). This, in itself, proves two things: that silencing is an active process led by the hermaphrodite and that oocyte licensing (some sort of mark or imprinting) clearly does not explain the findings that protected T sperm remain unprotected. These conclusions should be made clear in the text. Other support comes from figure S6e, where recovery from silencing after hrde mutation does not revert back even when wild type hrde1 is provided again. This further supports the conclusion that once you are protected and maternal transcripts deposited, they are passed on through generations and prevent activation of the piRNA pathways. This is important, in my view, since a big part of the paper hinges on the fact that although both protective and silencing signals appear to be RNA based: one of the signals comes from the maternally deposited homologous transcripts (explains the S5 results) and the other comes from an activation of piRNA dependent silencing. Parsing this out in the text is important to avoid confusing the reader.

2. The signal from the female is way stronger than that from the male, and this is a point that should be further highlighted in the text. Passage through the female is the key: again, further evidence for 1) signal is diffusible, 2) maternally deposited. These conclusions at the end of each sections would just reiterate the support for the main points.

Minor comments:

The section sub-headings should be revised to reflect the overall theme of the figures contained within them. This will help readers to navigate the concepts in the paper fluidly. I suggest that certain key terms added to the titles will help orient readers before they peruse each section.

Section 2: "Homologous maternal transcripts protect against initiation of mating induced silencing." The reason for this is that it helps readers to have consistent terminology and all through the paper the signal is called silencing or protection.

Section 4: "Other homologous sequences are trans silenced by signals generated from stable RNA silencing" This immediately tells the reader that this is a different phenomenon than stable silencing. Makes trans silencing the focus.

Section 5: Add the word trans...."Genes can recover from trans silencing....". separates the two phenomena better.

Some minor adjustment at the transitions between sections will work well with these headings so the reader can keep up.

Reviewer #3 (Remarks to the Author):

The revised version of Devanapally et al., which describes the silencing/expression behaviour of a transgene and modifications thereof, is a re-organised version of the manuscript that was originally submitted to Nature. Because of the COVID-19 crisis, unfortunately not much experimental work could be added to address the comments that had been given. The question is now whether this work, in its new shape, is a good fit for Nature Communications.

Let me start by stating that any opinion on this is obviously subjective. I state this, because the authors write in their rebuttal that my judgement on the significance of their work is subjective. I agree. I just wish to point out that also the two reviews that were more positive on the impact of their work, are subjective as well. There is no objective measure for judging how well papers fit to a given journal. I try to identify novel aspects in any manuscript I review, and then a) see how well they are supported by experiment and b) how novel they are.

In case of this particular manuscript, I think that very nice, and well executed experiments are presented, but that the impact of the results is limited. Looking at what Nature Communications typically publishes, I maintain my original view that more specialized journals, such as PLoS Genetics or EMBO Reports, are a more suitable publication venue for this work. I will walk through my original comments and the authors replies to them (in blue), and provide my responses in between.

Rebuttal text from authors:

We thank the reviewer for taking the time to review our manuscript. However, as pointed out by the other two reviewers our work is of broad significance as we shall elaborate in specific responses below.

I think that the general significance of this reduced by work that has been published in the past. Apart from possibly one exception, everything is compatible, and can be explained by models put forward in the past, in particular those on RNAe and RNAa. That one potential exception is that Tcherry can also protect gfp expressed from T, albeit at lower efficiency than the cherry that is expressed by T. However, this could be related to some low level of export of mRNAs in which cherry and gfp are still linked together. So, while this is a data point that may indeed challenge existing models, the presented data is too weak/limited to make a strong point out of it.

Rebuttal text from authors:

While this goal is of interest to researchers in the small RNA field, the results demonstrated in our manuscript are not as limited as characterized here. This is the reason we chose to document such comparisons only in a supplementary table (Supplementary Table 1).

The observations are mostly specific for T and derivatives thereof, and I do not see how the authors can maintain that their work has broad impact. All presented work point in the direction that T (and derivatives) is in a delicate balance when it comes to RNAe and RNAa. Most other transgenes are for some reason either much more inclined to RNAe (meaning stably silenced, and able to transmit this state onto other transgenes), and a few are strong in RNAa (meaning robustly expressed in the germline, and able to impose this activity on

normally silent transgenes). I do not claim that these two concepts have been well developed, even less that we fully understand them at the molecular level. However, the

current manuscript also does not provide such deeper understanding. It does provide a very useful, and well-described tool to study these concepts further.

I wish to express that these critiques are not meant to devalue the work. However, it may be harmful to the field to invoke new mechanisms without strong proof. The only really new feature here is that mating can induce the silencing, and as I already wrote in my first review, the effect of mating could just reflect absence of RNAa signals in the oocyte, leading to a flip to RNAe of T. No new concept is required to explain this. Hence, my view remains that the general impact of the work is limited.

Rebuttal text from authors:

Our work cannot be characterized as ‘descriptive and correlative’ because we repeatedly raise and test hypotheses. Below we provide a selection of experiments from the manuscript that sharpen and test hypotheses.

1. We distinguish between stochastic and deterministic reasons for escape from mating-induced silencing. We show that animals that are protected from or evade initiation of mating-induced silencing can undergo initiation in the next generation.

This is still descriptive, and does not provide novel insights. Good characteristic to know though for this system, in order to use it in experiments that will unveil what is happening.

2. We find a requirement for PRG-1, which is correlated with piRNA-mediated silencing. We then proceed to eliminate piRNA-binding sites to test this hypothesis and establish a role for piRNAs in silencing.

This is a nice experiment, but not very surprising. Again, this is a good characteristic to know though for this system, in order to use it in experiments that will unveil what is happening.

3. We provide strong evidence dismissing a common assumption in the field that piRNA-binding sites are predictive of transgenerational silencing. We show that transgenic mCherry sequence is silenced when inherited paternally but not when it is inherited maternally or when it is fused to an endogenous gene.

I do not contest the results, but I do contest the idea that these results dismiss the idea that piRNA binding sites can trigger RNAe. In fact the authors themselves show this for their system. piRNA binding definitely helps to predict silencing (this is not the same as: “always induce RNAe).

4. We dissect the gene sequence to find sequence elements that enable susceptibility to initiation (e.g. by deleting *unc-119(+)* or histone sequences, changing method of insertion or changing the genomic location) and that enable protection (e.g. by introducing point mutations into or truncating the maternal mCherry ORF).

This is nice, but it boils down to piRNA binding sites at the end. And that is not very novel. Other features cannot really be defined.

5. We provide evidence for maintenance of silencing being an active process that can span hundreds of generations rather than the possibility of permanent gene

inactivation by DNA mutation – first, by reactivating expression of inactive T upon hrde-1 removal and second, by observing the trans silencing effect of inactive T (even after >200 generations of being inactive) on other genes.

Reactivation and trans-silencing had been shown before, although I admit that this has not been explicitly published along with how long the respective strains had been in culture. So again: good characteristic to know though for this system, in order to use it in experiments that will unveil what is happening.

6. We test a requirement for sequence homology in trans silencing by deleting regions of homology on other genes.

This is nice, but it is rather detailed and specific, and does not provide the type of novel insights one would expect for Nature Comm.

7. A requirement for HRDE-1 in maintenance and initiation of silencing suggests small RNAs mediate transgenerational silencing of T and trans silencing of homologous genes. We use overexpression of GPR-1 as a tool to separate effects of maternal nuclear components (such as chromatin) from T to provide more evidence for this hypothesis and conclusively rule out chromatin-restricted factors.

This is indeed a nice experiment, but the outcome is not entirely novel. The GPR-1 approach is novel (and elegant!), but RNAe has been shown to transfer onto a male-incoming transgene without the maternal copy being there (Luteijn et al. 2012). This was simply done with heterozygous hermaphrodites, but the chromatin of the residual bodies do not come into contact with the male pro-nuclear contents anyway. The GPR-1 experiment presented here takes this further, and is a more robust test of any chromatin effect from the female, but the gain in insights is modest.

Rebuttal text from authors:

Two findings reported here transform our understanding of heredity by revealing unprecedented phenomena:

(1) We report the first stable transgenerational phenomenon (>300 generations) in any organism that can be easily initiated at will. [Previously reported phenomena like RNAe/paramutation are unpredictable and require generation of a new transgene (RNAe) or interaction with another transgene (paramutation) for silencing to be initiated each time];

This is not completely correct. RNAe can initiate on existing transgenes. One does not need to create them. I totally agree that this mating system is really cool, as it can be well-controlled, but I do not see the real novelty (yet).

Rebuttal text from authors:

(2) We provide evidence for transgenerational feedback mechanisms that enable recovery from silencing within the germline with gene context-specific kinetics. [More than a century of work has assumed that transgenerational epigenetic inheritance is limited by access to the germline - the so called 'Weismann Barrier'. Our results suggest that such an apparent barrier could be explained by transgenerational feedback

mechanisms that act within the germline at each gene.]

I agree that this is intriguing data, but thus far I'm afraid I'd still call that very descriptive. The link to Weissmann Barrier is very far-fetched.

The subjective judgement on the significance of our work made by this reviewer is therefore unsubstantiated. Claims of 'the authors fail to identify a new mechanism, and also do not uncover a totally new and unexpected phenomenon' contradict the views of the other two reviewers who recognize the unexpected results from mating-induced silencing, which this reviewer first described as 'an interesting phenomenon'.

As stated above, yes, my view is subjective.

In their rebuttal the authors next address each of my specific points. I paste these, including the authors responses below in italics. In between these points I enter my replies.

1) The authors say the GFP-Cherry transgene likely makes one fused transcript of the two genes, that are later decoupled through splicing. Given how central this transgene is to the story, this should simply be clearly resolved/demonstrated with experiments. smFISH shows separate puncta of mCherry and gfp mRNA in the cytosol, supporting separation of the two mRNAs through splicing. We discovered that the images also show one or two puncta within the nucleus that contain colocalized mCherry RNA and gfp RNA signals. These colocalized punctae are likely the sites of pre-mRNA transcription. We have now included these data (Fig. 1b, Supplementary Fig. 1b) as additional support.

This is nice additional data, although the resolution of the imaging is likely not sufficient to draw any conclusions on whether the overlapping nuclear signals represent non-spliced transcripts or simply two spliced transcripts that have not yet moved apart far enough to be able to see by imaging.

2) The simple fact that GFP and Cherry silencing not always go together cannot lead to the conclusion that two distinct mechanisms are acting on pre-mRNA and spliced mRNA. Given that the scoring of T activity is rather non-quantitative here, a simple threshold effect could also account for such a difference in silencing potency between GFP and Cherry dsRNA. Also the later claim that their experiments oppose the generality of chromatin induced silencing spreading over larger distances is a strong over-interpretation of a simple data-point.

This comment reflects a misunderstanding of the bases for our claims. We do not claim that distinct mechanisms are acting on pre-mRNA and spliced mRNA, nor do we use differences between gfp-dsRNA and mCherry-dsRNA to imply mechanistic differences. Rather, it is our claim that the observation of two generations of gfp mRNA silencing with unaffected mCherry expression upon gfp-dsRNA exposure deserves explanation. Prior expectations based on the spread of chromatin over long distances (Gu et al., Nature Genetics, 2012) and silencing at the pre-mRNA level by RNA-pol II stalling (Guang et al., Nature, 2010) is that both mCherry and gfp would be silenced. Our data is simply counter to this expectation. Our scoring of T activity in almost all cases was based upon imaging (we have > 10,000 images in this manuscript) and we categorized them based on rigorous quantification with ImageJ as presented in the manuscript. We also note that to dismiss a generality, a single well-done data point is sufficient. Nevertheless, in keeping with suggestions of other reviewers, we have removed this

section from the current submission.

Given that this has been removed it becomes irrelevant.

3) The mating induced silencing seems to be nothing more than that it reveals absence of maternal protection from silencing. The name implies much more, and is inappropriate.

Our many experiments finally established that mating-induced silencing occurs because of the absence of maternal protection. Of course, after we have established this explanation, one could characterize it as ‘nothing more than...’. In fact, the unexpected nature of this explanation is underscored by the absence of mating-induced silencing for many other tested genes. In other words, all these genes did not need any maternal protection. We strictly named the phenomenon based on how it arises without any assumptions about underlying ‘mechanisms’. Therefore, we maintain that mating induced silencing is a conservative and appropriate name for this phenomenon. As suggested by Referee 2, we have included explicit models supporting different conditions in Fig. 2 and Supplementary Figs. 1l and 5k.

I meant to convey that the name ‘mating-induced’ seems to reflect something special associated with mating. For instance, that the male sperm has some special feature, or that the act of mating induces some kind of signal allowing RNAe. Strictly speaking, I agree that the name is correct, but I am simply afraid that it may create the impression that another type of gene-silencing has been discovered, while that is not the case: it is fully compatible with already known RNAa-RNAe balance models, and this balance of course gets disrupted if the female does not have the homologous gene.

*4) The maternal protection mechanism could well be CSR-1 mediated, despite the claims made by the authors. The test with *cde-1* does not address this issue, as *cde-1* mutant have such a different phenotype than *csr-1*. It is clear that *cde-1* mutants do not recapitulate *csr-1* effects.*

*We shared this suspicion with the reviewer, which is why we did not claim that CSR-1 is not required and only noted that it is difficult to infer about the role of CSR-1 given pleiotropic effects caused by loss of *csr-1*. We also did not claim that *cde-1* loss*

*recapitulates *csr-1* loss. In fact we concluded ‘Thus, protection of T from mating-induced silencing relies on diffusible sequence-specific signals and could be independent of the CSR-1 pathway.’ [Underline added here for emphasis]. We have now reworded this section to make our caution clearer, including a statement acknowledging the need for additional experiments.*

The new text is better.

*5) The authors claim that re-silencing is actively established each generation. Isn’t this obvious? It would be naïve to assume that some non-active process could keep a transgene silent for 100s of generations. We also know that HRDE-1 is involved in that, so I am afraid I fail to see the why the result of reactivation by *hrde-1* mutation is so special.*

A transgene can remain ‘unexpressed’ for many reasons. The most obvious possibility with precedent is the acquisition of mutation (as occurs in repeat-induced point mutation

[RIP] in Neurospora), which is also triggered only during fertilization and depends on chromatin modifiers. Such mutational loss of expression would be a passive mechanism. We did not claim that reactivation upon loss of hrde-1 was special. Rather,

we used it to conclude that the loss of expression was due to active silencing and not passive mutation, especially because expression was not observed for hundreds of generations in 100% of the animals.

See above. If the goal was to show that T was not mutated, this should be stated as such. In that light it is a sort of control experiment. A good one, but not an experiment that brings a lot of novel insights.

6) Whether or not H3K9-methylation is involved has to be tested in wild-type versus hrde-1 mutants. No study in C. elegans claimed H3K9 methylation is causal and it could well be that T is H3K9 methylated also in active state. In fact, it may be THE reason it is so much prone to silencing.

Numerous studies have correlated H3K9 methylation with transgenerational stability of silencing (chiefly inspired by similar claims in the yeast chromatin field). Also see comments by Reviewer 1 “A key result in addition concerns the different

transgenerational silencing dynamics of two mRNAs in the same operon. The result suggests that transgenerational silencing can operate at the level of the mRNA and not upstream (chromatin or unspliced transcript). This goes counter to numerous previous reports in C. elegans that implicate histone modifications in the transgenerational transmission of silencing states.”.

Finally, it is a contradiction to claim both that H3K9 methylation is not causal and that ‘it may be THE reason it is so much prone to silencing’.

First, the final sentence of the above citation is not correct. A factor can be important in setting the conditions for silencing without itself driving the silencing.

Second, indeed, many studies have correlated H3K9 methylation with silencing. However, H3K9me3-modified chromatin can be expressed (see Buehler studies in pombe). My suggestion was that T may be H3K9me3 modified, while still being expressed, and that this status makes it prone to switch into RNAe. This would then be without an increase in H3K9me3, as it already has that. Hence my question was that the H3K9me3 status on T (and iT) should be address in wt vs hrde-1-/-, in addition to comparing T and iT (the latter is presented). I may well be wrong, but I think it is a reasonable hypothesis, and if true, one that would boost the impact of this work. The operon result that is referred to above (which has now been removed from the manuscript) does not really affect this reasoning.

7) The experiment with pgl-1 mutation, to test if protection of T is mediated by phase separation, is totally over interpreted. First, DEPS-1 is upstream of PGL-1, and DEPS-1 foci form in pgl-1 mutants. Second, so many indirect effects can be expected that the observed silencing could simply come from imbalances in the germline unrelated to phase separation directly.

We did not claim any specific role of phase separation. The reviewer appears to have mistaken our citation of previous work by others in the field as our claim. We quote the relevant section from our first submission here and note that the single mention of phase-separation is to cite the claims of others.

“The stable expression of T observed in the absence of mating suggests that transcripts from T engage protective mechanisms that have been proposed to ‘license’ expression within the germline²⁷. One such protective mechanism relies on phase-separated condensates within the germline called P-granules, which when disrupted can cause misregulation and aberrant

distribution of some transcripts^{28,29}. Consistent with P-granules facilitating stable expression of T, loss of the P-granule component PGL-1 resulted in

variable expression of T even in the absence of mating (Extended Data Fig. 8a). Therefore, the stable expression of T across generations within the hermaphrodite germline reflects reliable recognition of transcripts from T within P-granules as part of 'self' in every generation^{18,30,31}."

What we do claim is that the stable expression of T across generations reflects recognition of transcript from T as 'self' according to prevailing models.

The authors write in the current manuscript:

"This observation suggests that stable transgenerational expression of T likely reflects reliable recognition of transcripts from T within P granules as part of 'self' in every generation."

While carefully phrased, and disagree; at the very least it is a very much biased explanation. The observation may just as well suggest that in *pgl-1* mutants germline expression profiles are out of balance, in turn affecting T-status. Hence, I maintain my comment that this is over-interpretation.

Finally some other, smaller comments:

1) Introduction:

"Rare mutations in genome sequence that overcome DNA repair are transmitted across generations through DNA replication during each cell division."

Mutations are not recognized by DNA repair. Mismatches or damaged bases are recognized by DNA repair, and when DNA repair fails, they can give rise to a mutation.

2) Table S1: Luteijn et al. (2012) report on silencing that was sustained for >6 months. Even though generations were not explicitly counted, surely this would add up to >30 or so, presenting another case of very stable silencing.

3) "We noted that different *TcherryΔpi* variants appeared to protect in proportion to their coding-sequence lengths regardless of the number of mutated piRNA target sites (Figs. 2e, S5b and S5j). This observation suggests that these protective maternally inherited transcripts could be titrating away secondary small RNAs made against T triggered downstream of piRNA-binding in progeny (Figs. S5j and S5k)."

An alternative explanation could be that they (the *TcherryΔpi* variants) produce different amounts of protective 22G RNAs (most likely CSR-1 bound; however they may work).

4) "This trans silencing signal either is or relies on HRDE-1-dependnet small RNAs..."

Is that correct? I could also imagine another set of small RNAs that induce HRDE-1 to be loaded in the zygote with the respective 22G RNAs. I do not think the conclusion can be as strong as presented.

- 5) “Intriguingly, Tcherry Δ pi became resistant to trans silencing despite the continued presence of the silenced iT as evidenced by absence of nuclear-localized GFP and mCherry (compare iT/Tcherry vs. iT/Tcherry Δ pi in Fig. 5a).”

This sentence is confusing. The 'as evidenced' should reflect on the continued silencing of iT, but in this sentence it could also reflect on 'became resistant'. I had to read this sentence a number of times before it became clear to me what was meant. Please change.

- 6) "This observation suggests that piRNA binding to a target gene (Tcherry) or perfect homology to iT promotes its continued susceptibility to trans silencing by small RNAs made from iT."

The first option should be tested by crispering prg-1 in that strain (that takes out the piRNAs, without crossing).

- 7) The figures are still packed with numerous very small panels, that make this paper very hard to read. I think that is fine for a paper that provides many details on a specific subject, but it will prevent appreciation by a more broad audience.

In summary, my criticisms primarily target the novelty of the findings, not the quality of the work. I believe that this is a very thorough manuscript, that makes the T transgene into one of the best-described systems to study heritability of silencing. Interesting aspects are revealed that can really lead to papers that change the current views. The current manuscript, at least in my eyes, is an excellent preparatory study that sets the stage for such interesting follow-ups. I would recommend acceptance of this work for journals such as PLoS Genetics, EMBO reports, Genetics etc. However, in my perception Nat.Comm. has a different scope, and aims to publish manuscripts with a rather general impact, or studies in which the molecular understanding of a given process is significantly deepened, and this manuscript does not deliver that. Maternal protection against silencing, and that this depends on homologous genes expressed in the female germline, has been convincingly shown before (RNAa studies), and stable silencing has been demonstrated well in RNAe studies. The mating effect seems to be a logical consequence of these two concepts. The one results that does not fully match with these concepts (Tcherry protecting *gfp* expressed from T) has not been studied sufficiently to use it as a strong argument against these concepts. The authors may well be right, that this is that one data point that disproves a model, but for that it needs to be much better studied.

Reviewer #4 (Remarks to the Author):

In this manuscript Devanapally et al. describe an interesting new silencing phenomenon induced by mating that can persist for > 300 generations. The silencing is initiated by piRNAs and is maintained by the known downstream effectors of the piRNA pathway (the 22G-RNAs and the nuclear HRDE-1). The silencing of this specific single-copy transgene can also act in trans, even though for a certain number of generations. The same transgene silenced by dsRNAs is not continuously silenced as the mating-induced mechanism. All the experiments conducted are very rigorously done and the interpretation of their results is supported by the experimental evidence. I have not reviewed the first version of the manuscript, but it appears that the major concern raised by the reviewer #1 as well as the reviewer #2 was that the manuscript required extensive reorganization of the results and a better connection between the figures and the text. I can see that in this regard the manuscript has really improved and the presentation of the results now follow a much simpler logic. Nonetheless, it is still a quite dense paper in the way figures and results are shown and I think some additional changes and

clarification can further improve this very interesting study.

Major comments:

- The title states that "Mating can initiate stable RNA silencing that overcomes epigenetic recovery". However, this has been only shown on one type of single-copy transgenes (called T). Other MosSCI do not induce the mating silencing phenomenon (Supplementary Fig. 2). Thus, I wonder whether it would be more appropriate to state that : Mating can initiate stable RNA silencing of a single-copy transgene that overcomes epigenetic recovery.
- The abstract states that mating-induced silencing "is associated with small RNAs made in each generation". However, the authors have not measured (by sequencing or any other methods) small RNAs across generations. They indirectly deduced the production of small RNAs based on genetic experiments in Fig. 3. Therefore, it would be better to state that the mating-induced silencing requires components of the small RNA pathway or downstream component of the piRNA pathway or something like that.
- At the end of the introduction the authors claim that "particular cis regulatory sequences promote stable RNA silencing of genes expressed within the germline in *C. elegans*". It is still not clear to me which particular cis-regulatory sequences are required (see the comment below).
- To discover which parts of the T transgene are required for its susceptibility to mating-induced silencing the authors deleted many portions of the transgene and concluded that "operon structure, histone sequences, co-transformation marker (*C. briggsae* unc-119(+)), and the method of genomic integration are not sufficient to explain susceptibility to mating-induced silencing". They also show that changing the 3'UTR is not affecting the mating-induced silencing of Tcherry, suggesting that the 3'UTR is not sufficient to activate the mating-induced silencing. From all these experiments they concluded that "cis regulatory features (promoter, 3'UTR, 5'UTR, and introns in combination) of Tcherry are sufficient to change gene expression upon mating". However, it is still not clear to me how they arrive to such a conclusion. For instance, they have not changed introns and promoter. Therefore, it might be still possible that one of those elements might be sufficient to activate the mating-induced silencing. Moreover, in the experiment where they show that the same Cherry sequence fused to the endogenous mex-5 gene (by CRISPR-Cas9 I suppose) is not inducing mating-induced silencing, the possible explanation is that the endogenous maternal mex-5 mRNA can confer protection during mating (as shown later on in the paper). Also, why the other single-copy transgenes do not activate mating-induced silencing? Is it because they all contain a piece of endogenous mRNAs that can confer protection? This is why in my opinion it is still not clear which cis-regulatory elements are required to activate the mating-induced silencing.
- The figure 2D is difficult to follow without a schematic of the transgenes as shown in Supplementary Figure 3. I am wondering whether it would be possible to have the same schematic in Suppl. Fig. 3 and to indicate on a side whether it activates or not mating-induced silencing (a sort of scale with the schematic of the transgene). This type of representation will help to understand what they are changing in the T transgene. Same in other figures.
- Maybe it would be better to create one figure indicating only the genetic requirement of mating-induced silencing (the initiator and initiator/maintenance factors) and another figure (after it) showing the maternal protection from mating-induced silencing (including some key results from Supplementary figure 5).
- In figure 3 it is a little bit misleading to have in the title > 300 generation and in the figure > 150 generation.
- In Figure 4g it is very difficult to understand the sequence homology of the different transgenes compared to reference TCherry or TGFP. Maybe it is better to show some sort of percentage. Also, it is not clear to me why the alignment of Cherry and GFP from the reference T transgene does not give 100% of homology along the whole sequence (first lane in the two comparison).
- In Figure 4g it looks like that the trans silencing is just reducing the expression of the homologous GFP sequence (it looks like a maximum 2-fold reduction of GFP intensity) even though the gtbp::GFP sequence looks very similar (maybe 100%) to the above T GFP reference sequence. From the quantification of the GFP intensity, there are no silenced worms (with almost no GFP signal, meaning

silenced). I am wondering whether this can be called trans silencing or this is just a reduction in the level of GFP (still caused by small RNAs).

- Can the authors comment the results in figure 6 in light of the recent paper by Houri-Zeevi et al. about the three rules explaining the GFP RNAi inheritance? For instance, in fig. 6b why one line is more stably silenced than others even though they are all silenced in F2.

Minor comments:

- In the introduction *Cryptococcus neoformans* is misspelled.

- I am not sure the title of the figure 2 is explicative. Maybe is simpler to say genetic requirements of mating-induced RNA silencing or something like that?

- In figure 6b maybe they can label in the figure that the dsRNA is against the GFP from T transgene.

We thank the editor for getting our manuscript reviewed and for guidance in submitting this revised version. We thank the reviewers for their constructive comments. We have addressed their comments in this revised manuscript through modifications to the text and figures. Our specific responses are in blue below each reviewer comment. We hope that this improved manuscript now merits publication in Nature Communications.

Reviewer #2 (Remarks to the Author):

This much revised version of the Devanapally and Raman et al paper has improved greatly and focuses on mating induced silencing mechanism that the authors have found in *C. elegans*. The central finding is that mating can trigger silencing of non-self sequences via a piRNA mediated pathway. This silencing is remarkably stable >300 generations as opposed to other types of silencing mechanisms that only last for a few generations (~5-6). The paper is robust and thoughtful, and it has very impressive genetics (quality and quantity). In its current form, the concepts are easy to understand, the paper flows well, has a central topic and will definitely be an advance to the field. Although the *C. elegans* field is not a stranger to dsRNA silencing mechanisms and their transgenerational inheritance, this is a unique case where the initiation and maintenance of (and protection from) silencing are all derived from RNAs but in distinct pathways. This is enumerated in the paper through careful experimentation and reasoning. It highlights the versatility of RNA silencing mechanisms that can be used by an organism and has uncovered a reproducible and extremely strong method of transgenerational silencing that the authors hope will be used widely by the community for future studies. I strongly support the publication of this paper with a few changes made to the text, as follows.

We thank the reviewer for their positive words.

Major comment:

1. In the interest of highlighting the maternal contribution in the paper, it would serve the authors well to discuss that maternal transcripts provide both the silencing and the protection by pointing out the various lines of evidence that support that view. In the reasoning for figure 2: the authors point out that either oocyte licensing explains it all or that T (inherited through the sperm) remains protected in perpetuity. The key finding for this is that a protected cross progeny is still susceptible to silencing from mating with a mother that has never seen this sequence (Figs S5e and f). This, in itself, proves two things: that silencing is an active process led by the hermaphrodite and that oocyte licensing (some sort of mark or imprinting) clearly does not explain the findings that protected T sperm remain unprotected. These conclusions should be made clear in the text. Other support comes from figure S6e, where recovery from silencing after hrde mutation does not revert back even when wild type hrde1 is provided again. This further supports the conclusion that once you are protected and maternal transcripts deposited, they are passed on through generations and prevent activation of the piRNA pathways. This is important, in my view, since a big part of the paper hinges on the fact that although both protective and silencing signals appear to be RNA based: one of the signals comes from the maternally deposited homologous transcripts (explains the S5

results) and the other comes from an activation of piRNA dependent silencing. Parsing this out in the text is important to avoid confusing the reader.

We agree with the suggestion of the reviewer to pull together results that support maternal origin for both silencing and protective signals, and the results that reveal that the decision to silence is made in every generation regardless of past protection or is not remembered after disruption of silencing through loss of *hrde-1*. We have now included an early discussion paragraph that pulls together these findings from the paper and highlights that oocyte licensing, if present, is not permanent. The paragraph is pasted below for convenience:

Our analysis of the bicistronic operon *T* and its derivatives suggest that competing maternal signals establish gene expression in progeny. While maternally inherited PRG-1 and piRNAs mediate mating-induced silencing of the paternally inherited copy of *T* in progeny (Fig. 2), silencing is opposed whenever a maternal protective signal is present (Fig. 2f). This protective signal can act away from the maternal genome, and although its identity is currently unclear, two observations constrain possibilities. One, the maternal presence of part of the *mCherry* coding region from *T* can protect both *mCherry* and *gfp* expression (Fig. 2e), suggesting sequence-dependent recognition of unspliced pre-mRNA or DNA as the target to protect in cross progeny. Two, active *T* continues to be susceptible to mating-induced silencing regardless of protection or escape from silencing in previous generations (Supplementary Fig. 5e-f), suggesting that cross progeny need to inherit the maternal protective signal for consistent gene expression in every generation.

2.. The signal from the female is way stronger than that from the male, and this is a point that should be further highlighted in the text. Passage through the female is the key: again, further evidence for 1) signal is diffusible, 2) maternally deposited. These conclusions at the end of each sections would just reiterate the support for the main points.

We agree that these observations along with our *gpr-1* based experiments provide strong support for a diffusible signal that is maternally deposited. We have now reiterated these findings at the end of the corresponding results section.

Minor comments:

The section sub-headings should be revised to reflect the overall theme of the figures contained within them. This will help readers to navigate the concepts in the paper fluidly. I suggest that certain key terms added to the titles will help orient readers before they peruse each section.

We thank the reviewer for this suggestion and the specific guidance below.

Section 2: "Homologous maternal transcripts protect against initiation of mating induced silencing." The reason for this is that it helps readers to have consistent terminology and all through the paper the signal is called silencing or protection.

We have changed the section heading as suggested.

Section 4: "Other homologous sequences are trans silenced by signals generated from stable RNA silencing" This immediately tells the reader that this is a different phenomenon than stable silencing. Makes trans silencing the focus.

We have changed the title to read 'Signal associated with stable RNA silencing can enable trans silencing of homologous sequences' to keep the focus on trans silencing.

Section 5: Add the word trans...."Genes can recover from trans silencing....". separates the two phenomena better.

We have changed the title to read 'Genes can recover from and become resistant to trans silencing'

Some minor adjustment at the transitions between sections will work well with these headings so the reader can keep up.

We have adjusted the transition sentences to improve flow.

At the end of section 1:

In summary, mating can disrupt the expression of a set of single-copy transgenes and cause transgenerational RNA silencing. This stable RNA silencing could result from the activation/gain of mechanisms that promote silencing or the repression/loss of mechanisms that prevent silencing, or both.

At the end of section 2:

Thus, mating disrupts competing RNA-based mechanisms that regulate expression to initiate silencing (Fig. 2h) and maternal transcripts with partial homology are sufficient to oppose silencing by piRNAs. Protection by maternal transcripts explains the directionality of mating required for silencing and, in hindsight, also suggests explanations for the situations where we did not observe silencing (Supplementary Fig. 11).

At the end of section 4:

This difference in efficacy and/or transmission of the silencing signal could reflect differences in the intracellular environment in the two gametes and/or differences in the nature or levels of silencing signal inherited through the two gametes.

Reviewer #3 (Remarks to the Author):

We thank the reviewer for explaining their point of view in detail. While we do not agree with their perception of the level of novelty in our study and their classification of our experiments that test hypotheses as 'descriptive', we very much appreciate the detailed explanations and useful suggestions provided by this reviewer. Below, we first address the three comments on points in our rebuttal letter that require a response (comments 1, 6, and 7) and then address the 7 comments from this round of review.

1) Regarding smFISH supporting T being an operon:

This is nice additional data, although the resolution of the imaging is likely not sufficient to draw any conclusions on whether the overlapping nuclear signals represent non-spliced transcripts or simply two spliced transcripts that have not yet moved apart far enough to be able to see by imaging.

The overlapping smFISH signal for mCherry and for *gfp* within the nucleus is much brighter than the signal of single transcripts outside the nucleus. Therefore, the simplest interpretation is that these reflect the multiple nuclear transcripts localized at the site of transcription. While this observation is consistent with the possibility that *gfp* and mCherry are transcribed together as an operon, it does permit other possibilities such as simultaneous independent transcription. Therefore, we agree that additional experiments are needed to establish that *T* is indeed an operon. While the current version of the manuscript does not include the transgenerational effects on the operon in response to feeding RNAi, our results on protection of both *gfp* and *mCherry* from mating-induced silencing by fragments of *mCherry* sequence support this likely possibility of *oxSi487* encoding a two-gene operon as originally designed (Frøkjær-Jensen et al., 2012). Additional operon-related feeding RNAi experiments are beyond the scope of this study.

6) Regarding involvement of H3K9me3 as a reason for mating-induced silencing: First, the final sentence of the above citation is not correct. A factor can be important in setting the conditions for silencing without itself driving the silencing. Second, indeed, many studies have correlated H3K9 methylation with silencing. However, H3K9me3-modified chromatin can be expressed (see Buehler studies in *pombe*). My suggestion was that *T* may be H3K9me3 modified, while still being expressed, and that this status makes it prone to switch into RNAe. This would then be without an increase in H3K9me3, as it already has that. Hence my question was that the H3K9me3 status on *T* (and *iT*) should be address in *wt* vs *hrde-1-/-*, in addition to comparing *T* and *iT* (the latter is presented). I may well be wrong, but I think it is a reasonable hypothesis, and if true, one that would boost the impact of this work. The operon result that is referred to above (which has now been removed from the manuscript) does not really affect this reasoning.

Determining whether H3K9me3, even if present, is causal in any way requires many more experiments that are beyond the scope of this study. Future experiments, potentially through a genetic screen have the potential to identify the factors important for silencing in an unbiased manner. These screens could reveal H3K9me3, H3K27me3, and/or H3K23me3, which have all been reported downstream of HRDE-1 in the literature. Given that H3K23me3 was only recently discovered as a dominant modification generated in response to RNAi (Schwartz-Orbach et al., eLife, 2020), there could potentially be additional as yet unknown modifications.

7) Regarding loss of expression of *T* in *pgl-1(-)* animals:

The authors write in the current manuscript:

“This observation suggests that stable transgenerational expression of *T* likely

reflects reliable recognition of transcripts from T within P granules as part of 'self' in every generation."

While carefully phrased, and disagree; at the very least it is a very much biased explanation. The observation may just as well suggest that in *pgl-1* mutants germline expression profiles are out of balance, in turn affecting T-status. Hence, I maintain my comment that this is over-interpretation.

The designation of 'self' versus 'non-self' for transcripts that are ultimately expressed within an organism is problematic. How are we to know what an organism considers self? Nevertheless, many studies have espoused this nomenclature and we had included this sentence to provide context for comparison with these studies. We have now modified the sentence to read 'This observation suggests that stable transgenerational expression of T likely reflects reliable recognition of transcripts from T within P granules as part of 'self' in every generation, *according to some current models.*' [italics added here to highlight new text.]

Finally some other, smaller comments:

1) Introduction:

"Rare mutations in genome sequence that overcome DNA repair are transmitted across generations through DNA replication during each cell division."

Mutations are not recognized by DNA repair. Mismatches or damaged bases are recognized by DNA repair, and when DNA repair fails, they can give rise to a mutation.

We have reworded to say 'Rare mutations in genome sequence that result from failed DNA repair are transmitted across generations through DNA replication during each cell division.'

2) Table S1: Luteijn et al. (2012) report on silencing that was sustained for >6 months. Even though generations were not explicitly counted, surely this would add up to >30 or so, presenting another case of very stable silencing.

We have additionally included this citation in Supplementary Table 1, along with its previous inclusion in Supplementary Table 2 and throughout the manuscript.

3) "We noted that different T_{cherry}Δ_{pi} variants appeared to protect in proportion to their coding-sequence lengths regardless of the number of mutated piRNA target sites (Figs. 2e, S5b and S5j). This observation suggests that these protective maternally inherited transcripts could be titrating away secondary small RNAs made against T triggered downstream of piRNA-binding in progeny (Figs. S5j and S5k)."

An alternative explanation could be that they (the T_{cherry}Δ_{pi} variants) produce different amounts of protective 22G RNAs (most likely CSR-1 bound; however they may work).

We have now also included this alternative possibility.

4) “This trans silencing signal either is or relies on HRDE-1-dependent small RNAs...” Is that correct? I could also imagine another set of small RNAs that induce HRDE-1 to be loaded in the zygote with the respective 22G RNAs. I do not think the conclusion can be as strong as presented.

Our statement does not preclude additional classes of small RNAs that may also contribute to subsequent HRDE-1-dependent silencing (as indicated by ‘or relies on’ above). Therefore, the requirement for HRDE-1 and its associated RNAs remains.

5) “Intriguingly, *Tcherry*Δ*pi* became resistant to trans silencing despite the continued presence of the silenced *iT* as evidenced by absence of nuclear-localized GFP and mCherry (compare *iT/Tcherry* vs. *iT/Tcherry*Δ*pi* in Fig. 5a).”

This sentence is confusing. The ‘as evidenced’ should reflect on the continued silencing of *iT*, but in this sentence it could also reflect on ‘became resistant’. I had to read this sentence a number of times before it became clear to me what was meant. Please change.

We have reworded this into two sentences “Intriguingly, *Tcherry*Δ*pi* became resistant to trans silencing despite the continued presence of the silenced *iT* (compare *iT/Tcherry* vs. *iT/Tcherry*Δ*pi* in Fig. 5a). The continued silencing of *iT* was indicated by the absence of nuclear-localized GFP and mCherry.”

6) “This observation suggests that piRNA binding to a target gene (*Tcherry*) or perfect homology to *iT* promotes its continued susceptibility to trans silencing by small RNAs made from *iT*.”

The first option should be tested by crispering *prg-1* in that strain (that takes out the piRNAs, without crossing).

A recent preprint suggests that the role of PRG-1 is complex (Shukla et al., bioRxiv, 2021). They find that loss of PRG-1 promotes silencing by feeding RNAi through an unbiased screen for the enhanced maintenance of silencing. This finding based on PRG-1 is not consistent with one of the proposed possibilities based on piRNA-binding sites, i.e., that piRNAs could be needed to maintain *trans* silencing. Nevertheless, the final model proposed in Shukla et al., 2021 includes PRG-1 as being required for causing some silencing and for preventing other silencing, consistent with earlier work (de Albuquerque et al., Dev. Cell, 2015 and Phillips et al., Dev. Cell, 2015). Therefore, at this time more discerning experiments that potentially restrict PRG-1 activity in space and time are needed to clarify its role in silencing. We expect to develop tools in future studies that will enable such temporally and spatially restricted analyses.

7) The figures are still packed with numerous very small panels, that make this paper very hard to read. I think that is fine for a paper that provides many details on a specific subject, but it will prevent appreciation by a more broad audience.

We have added additional clarifications with the help of comments from reviewer #2 and reviewer #4 that we hope improve readability.

Reviewer #4 (Remarks to the Author):

In this manuscript Devanapally et al. describe an interesting new silencing phenomenon induced by mating that can persist for > 300 generations. The silencing is initiated by piRNAs and is maintained by the known downstream effectors of the piRNA pathway (the 22G-RNAs and the nuclear HRDE-1). The silencing of this specific single-copy transgene can also act in trans, even though for a certain number of generations. The same transgene silenced by dsRNAs is not continuously silenced as the mating-induced mechanism. All the experiments conducted are very rigorously done and the interpretation of their results is supported by the experimental evidence. I have not reviewed the first version of the manuscript, but it appears that the major concern raised by the reviewer #1 as well as the reviewer #2 was that the manuscript required extensive reorganization of the results and a better connection between the figures and the text. I can see that in this regard the manuscript has really improved and the presentation of the results now follow a much simpler logic. Nonetheless, it is still a quite dense paper in the way figures and results are shown and I think some additional changes and clarification can further improve this very interesting study.

We thank the reviewer for their positive words and the additional suggestions below for improving the manuscript.

Major comments:

- The title states that “Mating can initiate stable RNA silencing that overcomes epigenetic recovery”. However, this has been only shown on one type of single-copy transgenes (called T). Other MosSCI do not induce the mating silencing phenomenon (Supplementary Fig. 2). Thus, I wonder whether it would be more appropriate to state that : Mating can initiate stable RNA silencing of a single-copy transgene that overcomes epigenetic recovery.

We have observed mating-induced silencing in the transgene *T*, its derivatives generated through CRISPR modification of *T*, and its derivatives introduced at other loci through MosSCI and CRISPR. We agree that these transgenes could be considered as one type, however, we do not know yet what characteristics unite these transgenes and support such a classification. Our results demonstrate the possibility of single-copy sequences being subject to the phenomenon of mating-induced stable RNA silencing. Some of the numerous single-copy endogenous genes that undergo RNA silencing in every generation could have originated as mating-induced silencing and the endogenous gene *fem-1* can show silencing that is reminiscent of some aspects of mating-induced silencing, although its transgenerational stability cannot be evaluated. Therefore, we would like to retain the original title, but explicitly highlight in the abstract that this study uses a single-copy transgene and its derivatives to dissect this phenomenon. Specifically, we have stated:

Here we report that a minimal combination of *cis*-regulatory sequences can support permanent RNA silencing of a single-copy transgene and its derivatives in *C. elegans* simply upon mating.

- The abstract states that mating-induced silencing “is associated with small RNAs made in each generation”. However, the authors have not measured (by sequencing or any other methods) small RNAs across generations. They indirectly deduced the production of small RNAs based on genetic experiments in Fig. 3. Therefore, it would be better to state that the mating-induced silencing requires components of the small RNA pathway or downstream component of the piRNA pathway or something like that.

We have reworded the abstract to say that ‘This stable silencing requires components of the small RNA pathway and can silence homologous sequences in *trans*.’

- At the end of the introduction the authors claim that “particular *cis* regulatory sequences promote stable RNA silencing of genes expressed within the germline in *C. elegans*”. It is still not clear to me which particular *cis*-regulatory sequences are required (see the comment below).

We have rephrased the final paragraph of the introduction to be more specific: “Here we introduce mating as a simple approach to reproducibly initiate RNA silencing of a single-copy transgene that can last for hundreds of generations. A minimal combination of *cis*-regulatory sequences from this transgene can support such stable change within the *C. elegans* germline. Genes that share subsets of these regulatory sequences can be silenced for a few generations, but subsequently recover from and even become resistant to some forms of RNA silencing. Thus, our results establish a paradigm for analyzing the regulatory differences that determine persistent epigenetic change versus epigenetic recovery.”

- To discover which parts of the T transgene are required for its susceptibility to mating-induced silencing the authors deleted many portions of the transgene and concluded that “operon structure, histone sequences, co-transformation marker (*C. briggsae* unc-119(+)), and the method of genomic integration are not sufficient to explain susceptibility to mating-induced silencing”. They also show that changing the 3’UTR is not affecting the mating-induced silencing of Tcherry, suggesting that the 3’UTR is not sufficient to activate the mating-induced silencing. From all these experiments they concluded that “*cis* regulatory features (promoter, 3’UTR, 5’UTR, and introns in combination) of Tcherry are sufficient to change gene expression upon mating”. However, it is still not clear to me how they arrive to such a conclusion. For instance, they have not changed introns and promoter. Therefore, it might be still possible that one of those elements might be sufficient to activate the mating-induced silencing.

We thank the reviewer for pointing out this error. We did not highlight the basis for our inference that introns and promoters are not sufficient for supporting mating-induced silencing. Specifically, the introns or the same promoter being present at another MosSCI transgene did not cause mating-induced silencing. We have now highlighted

this result in the section preceding this conclusion sentence. We agree that we have not arrived at *the* minimal cis-regulatory element(s) that could be sufficient for activating mating-induced silencing. Future experiments could reveal that one or more sequence elements are sufficient for susceptibility to mating-induced silencing.

We have changed the concluding sentence of this result section to read “Thus, regulatory features that contribute to *Tcherry* expression (*cis* regulatory sequences, subnuclear localization of DNA, chromatin neighborhood, etc.) are sufficient to support change in gene expression upon mating.”

Moreover, in the experiment where they show that the same Cherry sequence fused to the endogenous *mex-5* gene (by CRISPR-Cas9 I suppose) is not inducing mating-induced silencing, the possible explanation is that the endogenous maternal *mex-5* mRNA can confer protection during mating (as shown later on in the paper). Also, why the other single-copy transgenes do not activate mating-induced silencing? Is it because they all contain a piece of endogenous mRNAs that can confer protection? This is why in my opinion it is still not clear which cis-regulatory elements are required to activate the mating-induced silencing.

The minimal sequence elements that allow silencing or prevent mating-induced silencing is currently unclear. In this study, we have eliminated many possibilities. Furthermore, sequence of *T* has endogenous histone mRNA sequences and yet shows mating-induced silencing despite the presence of multiple homologs of the histone gene (*his-58*, *his-66*, *his-48*) with $\geq 95\%$ sequence identity within the *C. elegans* genome, precluding the possibility that *any* endogenous mRNA piece could prevent mating-induced silencing. Other single-copy or even multi-copy transgenes that fail to show mating-induced silencing contain no shared endogenous sequence that can explain resistance to mating-induced silencing. Future studies dissecting both sequences that do not show mating-induced silencing and sequences that show mating-induced silencing could identify a smaller subset of features or a single feature that is sufficient for either property. We have therefore reworded the concluding paragraph in the introduction to avoid the possible interpretation that we have identified *the* minimal cis-regulatory feature(s).

- The figure 2D is difficult to follow without a schematic of the transgenes as shown in Supplementary Figure 3. I am wondering whether it would be possible to have the same schematic in Suppl. Fig. 3 and to indicate on a side whether it activates or not mating-induced silencing (a sort of scale with the schematic of the transgene). This type of representation will help to understand what they are changing in the T transgene. Same in other figures.

We have now included schematics as suggested in main figures that uses variants of *T* (Figure 2, Figure 4, and Figure 5). We have also added a column summarizing the susceptibility to mating-induced silencing of all transgenes listed in Supplementary Figure 3.

- Maybe it would be better to create one figure indicating only the genetic requirement of

mating-induced silencing (the initiator and initiator/maintenance factors) and another figure (after it) showing the maternal protection from mating-induced silencing (including some key results from Supplementary figure 5).

In our initial submission, we had used a presentation as suggested. However, reviewer #2 suggested that the mating-induced silencing be presented as initiation/protection and then maintenance. The reason for this presentation is to make it clear that maternal protection and initiation are both aspects of mating-induced silencing that occur in the initial generation, separate from maintenance.

- In figure 3 it is a little bit misleading to have in the title > 300 generation and in the figure > 150 generation.

We have clarified this by indicating in Figure 3b that the smFISH data was obtained using animals that have been silenced for 320 generations.

- In Figure 4g it is very difficult to understand the sequence homology of the different transgenes compared to reference TCherry or TGFP. Maybe it is better to show some sort of percentage. Also, it is not clear to me why the alignment of Cherry and GFP from the reference T transgene does not give 100% of homology along the whole sequence (first lane in the two comparison).

We thank the reviewer for pointing out this error. We have corrected this figure to depict pairwise alignments with *mCherry* and *gfp* sequences from *T* rather than the multiple sequence alignment depicted in the previous version. In addition to percentage identity, we have also highlighted the longest continuous stretches of homology because they could serve as templates for small RNA production or be targets of *trans* silencing. We have now also included all alignments as supplementary files for clarity.

- In Figure 4g it looks like that the *trans* silencing is just reducing the expression of the homologous GFP sequence (it looks like a maximum 2-fold reduction of GFP intensity) even though the *gtbp::GFP* sequence looks very similar (maybe 100%) to the above T GFP reference sequence. From the quantification of the GFP intensity, there are no silenced worms (with almost no GFP signal, meaning silenced). I am wondering whether this can be called *trans* silencing or this is just a reduction in the level of GFP (still caused by small RNAs).

Figure 4g reports measured fluorescence in absolute values, which includes the expression of GFP from *Pgtbp-1* not only in the germline, but also in somatic cells and from signal contributed by autofluorescence. As a result, a non-zero value is produced despite significant silencing within the germline. The silencing we observe within the germline is quite strong/robust as seen in the representative image shown in Supplemental Figure 7e. Thus, we expect that the quantification graph alone, without its accompanying image, provides an underestimate of *trans* silencing. Similarly, the quantification of silencing of *gtbp-1::mCherry* by *iT* is an underestimate of the silencing, which is also better observed in the images.

- Can the authors comment the results in figure 6 in light of the recent paper by Hour-Zeevi et al. about the three rules explaining the GFP RNAi inheritance? For instance, in fig. 6b why one line is more stably silenced than others even though they are all silenced in F2.

Our study has examined *gfp* silencing after expression using different *cis*-regulatory contexts within the *C. elegans* germline. In Hour-Zeevi et al. paper, feeding RNAi of a single *gfp* transgene is examined and partial correlations across generations are proposed as 'rules'. Therefore, we suspect that these rules are not general. In fact, the stochastic behavior of T in response to feeding RNAi can be seen after unbiased passaging even in the one line that shows TEI (Figure 6b in our manuscript). Similar stochasticity was also observed in response to neuronal dsRNA (as we showed in Devanapally et al., PNAS 2015, Suppl. Fig. 4). Our passaging across generations was using three L4-staged animals picked blindly and not by selecting single silenced worms. Therefore, it is also possible that passaging rare, partially silenced animals rather than passaging fully silenced animals contributed towards lineages with later apparent recovery of expression. However, even in Hour-Zeevi et al., there is evidence of recovery (see Figure 2E in Hour-Zeevi et al., which includes animals that switch from OFF to ON at F3 and F2 generations after P0 RNAi).

We had drawn attention to this relevant study through a brief note in the discussion "By comparing two different transgenes expressed within the germline, it was recently proposed that the duration of transgenerational silencing depends on stochastic 'states' adopted by individual organisms (Hour-Zeevi et al., 2020). However, examining the same coding sequences with different *cis*-regulatory sequences (Figure 6 and Supplementary Figure 2) suggests that the extent of silencing is not a cell or organism level property, but rather a gene level property."

Minor comments:

- In the introduction *Cryptococcus neoformans* is misspelled.

We have corrected the spelling.

- I am not sure the title of the figure 2 is explicative. Maybe is simpler to say genetic requirements of mating-induced RNA silencing or something like that?

Since this figure includes protection as well as genetic requirements for initiation, we have changed the title to read 'Requirements for initiation of and protection from mating-induced silencing'.

- In figure 6b maybe they can label in the figure that the dsRNA is against the GFP from T transgene.

We have included this information within the figure.

REVIEWERS' COMMENTS

Reviewer #4 (Remarks to the Author):

The Authors have addressed all my previous concerns. I therefore recommend their manuscript for publication.

We thank the editor for getting our manuscript reviewed and for guidance in submitting this revised version. We thank the reviewers for their constructive comments. We have addressed their comments in this revised manuscript through modifications to the text and figures. Our specific responses are in blue below each reviewer comment. We hope that this improved manuscript now merits publication in Nature Communications.

Reviewer #2 (Remarks to the Author):

This much revised version of the Devanapally and Raman et al paper has improved greatly and focuses on mating induced silencing mechanism that the authors have found in *C. elegans*. The central finding is that mating can trigger silencing of non-self sequences via a piRNA mediated pathway. This silencing is remarkably stable >300 generations as opposed to other types of silencing mechanisms that only last for a few generations (~5-6). The paper is robust and thoughtful, and it has very impressive genetics (quality and quantity). In its current form, the concepts are easy to understand, the paper flows well, has a central topic and will definitely be an advance to the field. Although the *C. elegans* field is not a stranger to dsRNA silencing mechanisms and their transgenerational inheritance, this is a unique case where the initiation and maintenance of (and protection from) silencing are all derived from RNAs but in distinct pathways. This is enumerated in the paper through careful experimentation and reasoning. It highlights the versatility of RNA silencing mechanisms that can be used by an organism and has uncovered a reproducible and extremely strong method of transgenerational silencing that the authors hope will be used widely by the community for future studies. I strongly support the publication of this paper with a few changes made to the text, as follows.

We thank the reviewer for their positive words.

Major comment:

1. In the interest of highlighting the maternal contribution in the paper, it would serve the authors well to discuss that maternal transcripts provide both the silencing and the protection by pointing out the various lines of evidence that support that view. In the reasoning for figure 2: the authors point out that either oocyte licensing explains it all or that T (inherited through the sperm) remains protected in perpetuity. The key finding for this is that a protected cross progeny is still susceptible to silencing from mating with a mother that has never seen this sequence (Figs S5e and f). This, in itself, proves two things: that silencing is an active process led by the hermaphrodite and that oocyte licensing (some sort of mark or imprinting) clearly does not explain the findings that protected T sperm remain unprotected. These conclusions should be made clear in the text. Other support comes from figure S6e, where recovery from silencing after hrde mutation does not revert back even when wild type hrde1 is provided again. This further supports the conclusion that once you are protected and maternal transcripts deposited, they are passed on through generations and prevent activation of the piRNA pathways. This is important, in my view, since a big part of the paper hinges on the fact that although both protective and silencing signals appear to be RNA based: one of the signals comes from the maternally deposited homologous transcripts (explains the S5

results) and the other comes from an activation of piRNA dependent silencing. Parsing this out in the text is important to avoid confusing the reader.

We agree with the suggestion of the reviewer to pull together results that support maternal origin for both silencing and protective signals, and the results that reveal that the decision to silence is made in every generation regardless of past protection or is not remembered after disruption of silencing through loss of *hrde-1*. We have now included an early discussion paragraph that pulls together these findings from the paper and highlights that oocyte licensing, if present, is not permanent. The paragraph is pasted below for convenience:

Our analysis of the bicistronic operon *T* and its derivatives suggest that competing maternal signals establish gene expression in progeny. While maternally inherited PRG-1 and piRNAs mediate mating-induced silencing of the paternally inherited copy of *T* in progeny (Fig. 2), silencing is opposed whenever a maternal protective signal is present (Fig. 2f). This protective signal can act away from the maternal genome, and although its identity is currently unclear, two observations constrain possibilities. One, the maternal presence of part of the *mCherry* coding region from *T* can protect both *mCherry* and *gfp* expression (Fig. 2e), suggesting sequence-dependent recognition of unspliced pre-mRNA or DNA as the target to protect in cross progeny. Two, active *T* continues to be susceptible to mating-induced silencing regardless of protection or escape from silencing in previous generations (Supplementary Fig. 5e-f), suggesting that cross progeny need to inherit the maternal protective signal for consistent gene expression in every generation.

2.. The signal from the female is way stronger than that from the male, and this is a point that should be further highlighted in the text. Passage through the female is the key: again, further evidence for 1) signal is diffusible, 2) maternally deposited. These conclusions at the end of each sections would just reiterate the support for the main points.

We agree that these observations along with our *gpr-1* based experiments provide strong support for a diffusible signal that is maternally deposited. We have now reiterated these findings at the end of the corresponding results section.

Minor comments:

The section sub-headings should be revised to reflect the overall theme of the figures contained within them. This will help readers to navigate the concepts in the paper fluidly. I suggest that certain key terms added to the titles will help orient readers before they peruse each section.

We thank the reviewer for this suggestion and the specific guidance below.

Section 2: "Homologous maternal transcripts protect against initiation of mating induced silencing." The reason for this is that it helps readers to have consistent terminology and all through the paper the signal is called silencing or protection.

We have changed the section heading as suggested.

Section 4: "Other homologous sequences are trans silenced by signals generated from stable RNA silencing" This immediately tells the reader that this is a different phenomenon than stable silencing. Makes trans silencing the focus.

We have changed the title to read 'Signal associated with stable RNA silencing can enable trans silencing of homologous sequences' to keep the focus on trans silencing.

Section 5: Add the word trans...."Genes can recover from trans silencing....". separates the two phenomena better.

We have changed the title to read 'Genes can recover from and become resistant to trans silencing'

Some minor adjustment at the transitions between sections will work well with these headings so the reader can keep up.

We have adjusted the transition sentences to improve flow.

At the end of section 1:

In summary, mating can disrupt the expression of a set of single-copy transgenes and cause transgenerational RNA silencing. This stable RNA silencing could result from the activation/gain of mechanisms that promote silencing or the repression/loss of mechanisms that prevent silencing, or both.

At the end of section 2:

Thus, mating disrupts competing RNA-based mechanisms that regulate expression to initiate silencing (Fig. 2h) and maternal transcripts with partial homology are sufficient to oppose silencing by piRNAs. Protection by maternal transcripts explains the directionality of mating required for silencing and, in hindsight, also suggests explanations for the situations where we did not observe silencing (Supplementary Fig. 11).

At the end of section 4:

This difference in efficacy and/or transmission of the silencing signal could reflect differences in the intracellular environment in the two gametes and/or differences in the nature or levels of silencing signal inherited through the two gametes.

Reviewer #3 (Remarks to the Author):

We thank the reviewer for explaining their point of view in detail. While we do not agree with their perception of the level of novelty in our study and their classification of our experiments that test hypotheses as 'descriptive', we very much appreciate the detailed explanations and useful suggestions provided by this reviewer. Below, we first address the three comments on points in our rebuttal letter that require a response (comments 1, 6, and 7) and then address the 7 comments from this round of review.

1) Regarding smFISH supporting T being an operon:

This is nice additional data, although the resolution of the imaging is likely not sufficient to draw any conclusions on whether the overlapping nuclear signals represent non-spliced transcripts or simply two spliced transcripts that have not yet moved apart far enough to be able to see by imaging.

The overlapping smFISH signal for mCherry and for *gfp* within the nucleus is much brighter than the signal of single transcripts outside the nucleus. Therefore, the simplest interpretation is that these reflect the multiple nuclear transcripts localized at the site of transcription. While this observation is consistent with the possibility that *gfp* and mCherry are transcribed together as an operon, it does permit other possibilities such as simultaneous independent transcription. Therefore, we agree that additional experiments are needed to establish that *T* is indeed an operon. While the current version of the manuscript does not include the transgenerational effects on the operon in response to feeding RNAi, our results on protection of both *gfp* and *mCherry* from mating-induced silencing by fragments of *mCherry* sequence support this likely possibility of *oxSi487* encoding a two-gene operon as originally designed (Frøkjær-Jensen et al., 2012). Additional operon-related feeding RNAi experiments are beyond the scope of this study.

6) Regarding involvement of H3K9me3 as a reason for mating-induced silencing: First, the final sentence of the above citation is not correct. A factor can be important in setting the conditions for silencing without itself driving the silencing. Second, indeed, many studies have correlated H3K9 methylation with silencing. However, H3K9me3-modified chromatin can be expressed (see Buehler studies in *pombe*). My suggestion was that *T* may be H3K9me3 modified, while still being expressed, and that this status makes it prone to switch into RNAe. This would then be without an increase in H3K9me3, as it already has that. Hence my question was that the H3K9me3 status on *T* (and *iT*) should be address in *wt vs hrde-1-/-*, in addition to comparing *T* and *iT* (the latter is presented). I may well be wrong, but I think it is a reasonable hypothesis, and if true, one that would boost the impact of this work. The operon result that is referred to above (which has now been removed from the manuscript) does not really affect this reasoning.

Determining whether H3K9me3, even if present, is causal in any way requires many more experiments that are beyond the scope of this study. Future experiments, potentially through a genetic screen have the potential to identify the factors important for silencing in an unbiased manner. These screens could reveal H3K9me3, H3K27me3, and/or H3K23me3, which have all been reported downstream of HRDE-1 in the literature. Given that H3K23me3 was only recently discovered as a dominant modification generated in response to RNAi (Schwartz-Orbach et al., eLife, 2020), there could potentially be additional as yet unknown modifications.

7) Regarding loss of expression of *T* in *pgl-1(-)* animals:

The authors write in the current manuscript:

“This observation suggests that stable transgenerational expression of *T* likely

reflects reliable recognition of transcripts from T within P granules as part of 'self' in every generation."

While carefully phrased, and disagree; at the very least it is a very much biased explanation. The observation may just as well suggest that in *pgl-1* mutants germline expression profiles are out of balance, in turn affecting T-status. Hence, I maintain my comment that this is over-interpretation.

The designation of 'self' versus 'non-self' for transcripts that are ultimately expressed within an organism is problematic. How are we to know what an organism considers self? Nevertheless, many studies have espoused this nomenclature and we had included this sentence to provide context for comparison with these studies. We have now modified the sentence to read 'This observation suggests that stable transgenerational expression of T likely reflects reliable recognition of transcripts from T within P granules as part of 'self' in every generation, *according to some current models.*' [italics added here to highlight new text.]

Finally some other, smaller comments:

1) Introduction:

"Rare mutations in genome sequence that overcome DNA repair are transmitted across generations through DNA replication during each cell division."

Mutations are not recognized by DNA repair. Mismatches or damaged bases are recognized by DNA repair, and when DNA repair fails, they can give rise to a mutation.

We have reworded to say 'Rare mutations in genome sequence that result from failed DNA repair are transmitted across generations through DNA replication during each cell division.'

2) Table S1: Luteijn et al. (2012) report on silencing that was sustained for >6 months. Even though generations were not explicitly counted, surely this would add up to >30 or so, presenting another case of very stable silencing.

We have additionally included this citation in Supplementary Table 1, along with its previous inclusion in Supplementary Table 2 and throughout the manuscript.

3) "We noted that different Tcherry Δ pi variants appeared to protect in proportion to their coding-sequence lengths regardless of the number of mutated piRNA target sites (Figs. 2e, S5b and S5j). This observation suggests that these protective maternally inherited transcripts could be titrating away secondary small RNAs made against T triggered downstream of piRNA-binding in progeny (Figs. S5j and S5k)."

An alternative explanation could be that they (the Tcherry Δ pi variants) produce different amounts of protective 22G RNAs (most likely CSR-1 bound; however they may work).

We have now also included this alternative possibility.

4) “This trans silencing signal either is or relies on HRDE-1-dependent small RNAs...” Is that correct? I could also imagine another set of small RNAs that induce HRDE-1 to be loaded in the zygote with the respective 22G RNAs. I do not think the conclusion can be as strong as presented.

Our statement does not preclude additional classes of small RNAs that may also contribute to subsequent HRDE-1-dependent silencing (as indicated by ‘or relies on’ above). Therefore, the requirement for HRDE-1 and its associated RNAs remains.

5) “Intriguingly, *TcherryΔpi* became resistant to trans silencing despite the continued presence of the silenced *iT* as evidenced by absence of nuclear-localized GFP and mCherry (compare *iT/Tcherry* vs. *iT/TcherryΔpi* in Fig. 5a).” This sentence is confusing. The ‘as evidenced’ should reflect on the continued silencing of *iT*, but in this sentence it could also reflect on ‘became resistant’. I had to read this sentence a number of times before it became clear to me what was meant. Please change.

We have reworded this into two sentences “Intriguingly, *TcherryΔpi* became resistant to trans silencing despite the continued presence of the silenced *iT* (compare *iT/Tcherry* vs. *iT/TcherryΔpi* in Fig. 5a). The continued silencing of *iT* was indicated by the absence of nuclear-localized GFP and mCherry.”

6) “This observation suggests that piRNA binding to a target gene (*Tcherry*) or perfect homology to *iT* promotes its continued susceptibility to trans silencing by small RNAs made from *iT*.”

The first option should be tested by crispering *prg-1* in that strain (that takes out the piRNAs, without crossing).

A recent preprint suggests that the role of PRG-1 is complex (Shukla et al., bioRxiv, 2021). They find that loss of PRG-1 promotes silencing by feeding RNAi through an unbiased screen for the enhanced maintenance of silencing. This finding based on PRG-1 is not consistent with one of the proposed possibilities based on piRNA-binding sites, i.e., that piRNAs could be needed to maintain *trans* silencing. Nevertheless, the final model proposed in Shukla et al., 2021 includes PRG-1 as being required for causing some silencing and for preventing other silencing, consistent with earlier work (de Albuquerque et al., Dev. Cell, 2015 and Phillips et al., Dev. Cell, 2015). Therefore, at this time more discerning experiments that potentially restrict PRG-1 activity in space and time are needed to clarify its role in silencing. We expect to develop tools in future studies that will enable such temporally and spatially restricted analyses.

7) The figures are still packed with numerous very small panels, that make this paper very hard to read. I think that is fine for a paper that provides many details on a specific subject, but it will prevent appreciation by a more broad audience.

We have added additional clarifications with the help of comments from reviewer #2 and reviewer #4 that we hope improve readability.

Reviewer #4 (Remarks to the Author):

In this manuscript Devanapally et al. describe an interesting new silencing phenomenon induced by mating that can persist for > 300 generations. The silencing is initiated by piRNAs and is maintained by the known downstream effectors of the piRNA pathway (the 22G-RNAs and the nuclear HRDE-1). The silencing of this specific single-copy transgene can also act in trans, even though for a certain number of generations. The same transgene silenced by dsRNAs is not continuously silenced as the mating-induced mechanism. All the experiments conducted are very rigorously done and the interpretation of their results is supported by the experimental evidence. I have not reviewed the first version of the manuscript, but it appears that the major concern raised by the reviewer #1 as well as the reviewer #2 was that the manuscript required extensive reorganization of the results and a better connection between the figures and the text. I can see that in this regard the manuscript has really improved and the presentation of the results now follow a much simpler logic. Nonetheless, it is still a quite dense paper in the way figures and results are shown and I think some additional changes and clarification can further improve this very interesting study.

We thank the reviewer for their positive words and the additional suggestions below for improving the manuscript.

Major comments:

- The title states that “Mating can initiate stable RNA silencing that overcomes epigenetic recovery”. However, this has been only shown on one type of single-copy transgenes (called T). Other MosSCI do not induce the mating silencing phenomenon (Supplementary Fig. 2). Thus, I wonder whether it would be more appropriate to state that : Mating can initiate stable RNA silencing of a single-copy transgene that overcomes epigenetic recovery.

We have observed mating-induced silencing in the transgene *T*, its derivatives generated through CRISPR modification of *T*, and its derivatives introduced at other loci through MosSCI and CRISPR. We agree that these transgenes could be considered as one type, however, we do not know yet what characteristics unite these transgenes and support such a classification. Our results demonstrate the possibility of single-copy sequences being subject to the phenomenon of mating-induced stable RNA silencing. Some of the numerous single-copy endogenous genes that undergo RNA silencing in every generation could have originated as mating-induced silencing and the endogenous gene *fem-1* can show silencing that is reminiscent of some aspects of mating-induced silencing, although its transgenerational stability cannot be evaluated. Therefore, we would like to retain the original title, but explicitly highlight in the abstract that this study uses a single-copy transgene and its derivatives to dissect this phenomenon. Specifically, we have stated:

Here we report that a minimal combination of *cis*-regulatory sequences can support permanent RNA silencing of a single-copy transgene and its derivatives in *C. elegans* simply upon mating.

- The abstract states that mating-induced silencing “is associated with small RNAs made in each generation”. However, the authors have not measured (by sequencing or any other methods) small RNAs across generations. They indirectly deduced the production of small RNAs based on genetic experiments in Fig. 3. Therefore, it would be better to state that the mating-induced silencing requires components of the small RNA pathway or downstream component of the piRNA pathway or something like that.

We have reworded the abstract to say that ‘This stable silencing requires components of the small RNA pathway and can silence homologous sequences in *trans*.’

- At the end of the introduction the authors claim that “particular *cis* regulatory sequences promote stable RNA silencing of genes expressed within the germline in *C. elegans*”. It is still not clear to me which particular *cis*-regulatory sequences are required (see the comment below).

We have rephrased the final paragraph of the introduction to be more specific: “Here we introduce mating as a simple approach to reproducibly initiate RNA silencing of a single-copy transgene that can last for hundreds of generations. A minimal combination of *cis*-regulatory sequences from this transgene can support such stable change within the *C. elegans* germline. Genes that share subsets of these regulatory sequences can be silenced for a few generations, but subsequently recover from and even become resistant to some forms of RNA silencing. Thus, our results establish a paradigm for analyzing the regulatory differences that determine persistent epigenetic change versus epigenetic recovery.”

- To discover which parts of the T transgene are required for its susceptibility to mating-induced silencing the authors deleted many portions of the transgene and concluded that “operon structure, histone sequences, co-transformation marker (*C. briggsae* unc-119(+)), and the method of genomic integration are not sufficient to explain susceptibility to mating-induced silencing”. They also show that changing the 3’UTR is not affecting the mating-induced silencing of Tcherry, suggesting that the 3’UTR is not sufficient to activate the mating-induced silencing. From all these experiments they concluded that “*cis* regulatory features (promoter, 3’UTR, 5’UTR, and introns in combination) of Tcherry are sufficient to change gene expression upon mating”. However, it is still not clear to me how they arrive to such a conclusion. For instance, they have not changed introns and promoter. Therefore, it might be still possible that one of those elements might be sufficient to activate the mating-induced silencing.

We thank the reviewer for pointing out this error. We did not highlight the basis for our inference that introns and promoters are not sufficient for supporting mating-induced silencing. Specifically, the introns or the same promoter being present at another MosSCI transgene did not cause mating-induced silencing. We have now highlighted

this result in the section preceding this conclusion sentence. We agree that we have not arrived at *the* minimal cis-regulatory element(s) that could be sufficient for activating mating-induced silencing. Future experiments could reveal that one or more sequence elements are sufficient for susceptibility to mating-induced silencing.

We have changed the concluding sentence of this result section to read “Thus, regulatory features that contribute to *Tcherry* expression (*cis* regulatory sequences, subnuclear localization of DNA, chromatin neighborhood, etc.) are sufficient to support change in gene expression upon mating.”

Moreover, in the experiment where they show that the same Cherry sequence fused to the endogenous *mex-5* gene (by CRISPR-Cas9 I suppose) is not inducing mating-induced silencing, the possible explanation is that the endogenous maternal *mex-5* mRNA can confer protection during mating (as shown later on in the paper). Also, why the other single-copy transgenes do not activate mating-induced silencing? Is it because they all contain a piece of endogenous mRNAs that can confer protection? This is why in my opinion it is still not clear which cis-regulatory elements are required to activate the mating-induced silencing.

The minimal sequence elements that allow silencing or prevent mating-induced silencing is currently unclear. In this study, we have eliminated many possibilities. Furthermore, sequence of *T* has endogenous histone mRNA sequences and yet shows mating-induced silencing despite the presence of multiple homologs of the histone gene (*his-58*, *his-66*, *his-48*) with $\geq 95\%$ sequence identity within the *C. elegans* genome, precluding the possibility that *any* endogenous mRNA piece could prevent mating-induced silencing. Other single-copy or even multi-copy transgenes that fail to show mating-induced silencing contain no shared endogenous sequence that can explain resistance to mating-induced silencing. Future studies dissecting both sequences that do not show mating-induced silencing and sequences that show mating-induced silencing could identify a smaller subset of features or a single feature that is sufficient for either property. We have therefore reworded the concluding paragraph in the introduction to avoid the possible interpretation that we have identified *the* minimal cis-regulatory feature(s).

- The figure 2D is difficult to follow without a schematic of the transgenes as shown in Supplementary Figure 3. I am wondering whether it would be possible to have the same schematic in Suppl. Fig. 3 and to indicate on a side whether it activates or not mating-induced silencing (a sort of table with the schematic of the transgene). This type of representation will help to understand what they are changing in the T transgene. Same in other figures.

We have now included schematics as suggested in main figures that uses variants of *T* (Figure 2, Figure 4, and Figure 5). We have also added a column summarizing the susceptibility to mating-induced silencing of all transgenes listed in Supplementary Figure 3.

- Maybe it would be better to create one figure indicating only the genetic requirement of

mating-induced silencing (the initiator and initiator/maintenance factors) and another figure (after it) showing the maternal protection from mating-induced silencing (including some key results from Supplementary figure 5).

In our initial submission, we had used a presentation as suggested. However, reviewer #2 suggested that the mating-induced silencing be presented as initiation/protection and then maintenance. The reason for this presentation is to make it clear that maternal protection and initiation are both aspects of mating-induced silencing that occur in the initial generation, separate from maintenance.

- In figure 3 it is a little bit misleading to have in the title > 300 generation and in the figure > 150 generation.

We have clarified this by indicating in Figure 3b that the smFISH data was obtained using animals that have been silenced for 320 generations.

- In Figure 4g it is very difficult to understand the sequence homology of the different transgenes compared to reference TCherry or TGFP. Maybe it is better to show some sort of percentage. Also, it is not clear to me why the alignment of Cherry and GFP from the reference T transgene does not give 100% of homology along the whole sequence (first lane in the two comparison).

We thank the reviewer for pointing out this error. We have corrected this figure to depict pairwise alignments with *mCherry* and *gfp* sequences from *T* rather than the multiple sequence alignment depicted in the previous version. In addition to percentage identity, we have also highlighted the longest continuous stretches of homology because they could serve as templates for small RNA production or be targets of *trans* silencing. We have now also included all alignments as supplementary files for clarity.

- In Figure 4g it looks like that the trans silencing is just reducing the expression of the homologous GFP sequence (it looks like a maximum 2-fold reduction of GFP intensity) even though the *gtbp::GFP* sequence looks very similar (maybe 100%) to the above T GFP reference sequence. From the quantification of the GFP intensity, there are no silenced worms (with almost no GFP signal, meaning silenced). I am wondering whether this can be called trans silencing or this is just a reduction in the level of GFP (still caused by small RNAs).

Figure 4g reports measured fluorescence in absolute values, which includes the expression of GFP from *Pgtbp-1* not only in the germline, but also in somatic cells and from signal contributed by autofluorescence. As a result, a non-zero value is produced despite significant silencing within the germline. The silencing we observe within the germline is quite strong/robust as seen in the representative image shown in Supplemental Figure 7e. Thus, we expect that the quantification graph alone, without its accompanying image, provides an underestimate of *trans* silencing. Similarly, the quantification of silencing of *gtbp-1::mCherry* by *iT* is an underestimate of the silencing, which is also better observed in the images.

- Can the authors comment the results in figure 6 in light of the recent paper by Hourii-Zeevi et al. about the three rules explaining the GFP RNAi inheritance? For instance, in fig. 6b why one line is more stably silenced than others even though they are all silenced in F2.

Our study has examined *gfp* silencing after expression using different *cis*-regulatory contexts within the *C. elegans* germline. In the Hourii-Zeevi et al. paper, feeding RNAi of a single *gfp* transgene is examined and partial correlations across generations are proposed as 'rules'. Therefore, we suspect that these rules are not general. In fact, the stochastic behavior of T in response to feeding RNAi can be seen after unbiased passaging even in the one line that shows TEI (Figure 6b in our manuscript). Similar stochasticity was also observed in response to neuronal dsRNA (as we showed in Devanapally et al., PNAS 2015, Suppl. Fig. 4). Our passaging across generations was using three L4-staged animals picked blindly and not by selecting single silenced worms. Therefore, it is also possible that passaging rare, partially silenced animals rather than passaging fully silenced animals contributed towards lineages with later apparent recovery of expression. However, even in Hourii-Zeevi et al., there is evidence of recovery (see Figure 2E in Hourii-Zeevi et al., which includes animals that switch from OFF to ON at F3 and F2 generations after P0 RNAi).

We had drawn attention to this relevant study through a brief note in the discussion "By comparing two different transgenes expressed within the germline, it was recently proposed that the duration of transgenerational silencing depends on stochastic 'states' adopted by individual organisms (Hourii-Zeevi et al., 2020). However, examining the same coding sequences with different *cis*-regulatory sequences (Figure 6 and Supplementary Figure 2) suggests that the extent of silencing is not a cell or organism level property, but rather a gene level property."

Minor comments:

- In the introduction *Cryptococcus neoformans* is misspelled.

We have corrected the spelling.

- I am not sure the title of the figure 2 is explicative. Maybe is simpler to say genetic requirements of mating-induced RNA silencing or something like that?

Since this figure includes protection as well as genetic requirements for initiation, we have changed the title to read 'Requirements for initiation of and protection from mating-induced silencing'.

- In figure 6b maybe they can label in the figure that the dsRNA is against the GFP from T transgene.

We have included this information within the figure.